# Reprogramming Pretrained Target-Specific Diffusion Models for Dual-Target Drug Design

**Xiangxin Zhou**[1,2]          **Jiaqi Guan**[3]          **Yijia Zhang**[4]

**Xingang Peng**[5]          **Liang Wang**[1,2]          **Jianzhu Ma**[4,6,*]

[1]School of Artificial Intelligence, University of Chinese Academy of Sciences
[2]New Laboratory of Pattern Recognition (NLPR),
State Key Laboratory of Multimodal Artificial Intelligence Systems (MAIS),
Institute of Automation, Chinese Academy of Sciences (CASIA)
[3]Department of Computer Science, University of Illinois Urbana-Champaign
[4]Department of Electronic Engineering, Tsinghua University
[5]Institute for Artificial Intelligence, Peking University
[6]Institute for AI Industry Research, Tsinghua University

## Abstract

Dual-target therapeutic strategies have become a compelling approach and attracted significant attention due to various benefits, such as their potential in overcoming drug resistance in cancer therapy. Considering the tremendous success that deep generative models have achieved in structure-based drug design in recent years, we formulate dual-target drug design as a generative task and curate a novel dataset of potential target pairs based on synergistic drug combinations. We propose to design dual-target drugs with diffusion models that are trained on single-target protein-ligand complex pairs. Specifically, we align two pockets in 3D space with protein-ligand binding priors and build two complex graphs with shared ligand nodes for SE(3)-equivariant composed message passing, based on which we derive a composed drift in both 3D and categorical probability space in the generative process. Our algorithm can well transfer the knowledge gained in single-target pretraining to dual-target scenarios in a zero-shot manner. We also repurpose linker design methods as strong baselines for this task. Extensive experiments demonstrate the effectiveness of our method compared with various baselines.

## 1   Introduction

A promising paradigm of rational drug design is structure-based drug design (SBDD) [1], which uses computational chemistry tools in which the 3D structure of a protein target is used as the basis to identify or design new chemical entities. The foundation of structure-based drug design has been grounded in the lock-and-key hypothesis, positing that an optimal ligand molecule should possess a structure that is complementary to the target site. Recently, dual-target drug design, which aims to design "one key" for "two locks", has attracted significant attention. Precisely, dual-target drug design [4] is a strategy in pharmaceutical research that aims to design single ligand molecules capable of interacting with two different biological targets simultaneously. A dual-target drug can potentially lower the odds of resistance developing [62] and effectively manage the disease which involve complex biological pathways with multiple proteins [49]. Recent years have witnessed a noticeable

---

*Correspondence to: Jianzhu Ma <majianzhu@tsinghua.edu.cn>.

38th Conference on Neural Information Processing Systems (NeurIPS 2024).

increase in the FDA's approval of dual-target drugs [32, 34]. Please refer to Appendix A for a more comprehensive understanding of motivation, significance, and current practices of dual-target drug design.

Deep learning, particularly deep generative models [63] and geometric deep learning [46], has been introduced to SBDD and achieved promising results. Peng et al. [45], Zhang et al. [65] proposed to sequentially generate atoms or fragments using auto-regressive generative models conditioned on a specific protein binding site. Guan et al. [19], Lin et al. [33], Schneuing et al. [53] proposed to generate ligand molecules with diffusion models and achieved high binding affinity. However, due to the scarcity of data resources and high computational complexity, there is limited progress on introducing powerful generative models into dual-target drug design. Besides, there also lacks a comprehensive benchmark and dataset for evaluating the dual-target drug design, which also hinders the community from developing AI-powered computational tools for dual-target drug design.

To overcome the aforementioned challenges, we first propose a dataset for dual-target drug design. The design of dual-target drugs for arbitrary target pairs lacks substantive purpose. Inspired by the concept of drug synergism [58], where the combined effect of two drugs surpasses the effects of each drug when used individually, we carefully select pairs of targets from combinations of drugs that demonstrate significant synergistic interactions. The effectiveness of such combination therapy [39, 50, 44] has demonstrate significant efficacy in tumor eradication at both cellular level and in vivo study. Designing dual-target drugs for the paired targets may further improve the efficacy and reduce side effects. Additionally, we also provide a reference ligand for each target and the 3D structure of each protein-ligand complex in our dataset. Besides, we formulate the dual-target drug design as a generative task, based on which we further propose to reprogram pretrained target-specific diffusion models as introduced by Guan et al. [19] for the dual-target setting in zero-shot manner. More specifically, we first align dual pockets in 3D space with protein-ligand interaction priors that encapsulate the intricate features of the pockets. We compose the predicted drift terms in both 3D and categorical probability space in the reverse generative process of the diffusion model to generate dual-target drugs. We name this method as COMPDIFF. We further improve this method by building two complex graphs with shared ligand nodes for SE(3)-equivariant composed message passing. In this method, we compose the SE(3)-equivariant message at each layer of the equivariant neural network instead of only on the output level. We name this method as DUALDIFF. Our approach effectively transfers the knowledge acquired from pretraining on single-target datasets, circumventing the challenging demand for extensive training data required for dual-target drug design. We also repurpose linker design methods [22, 17] as a strong baseline for this task. We outline strategies to identify potential fragments from the synergistic drug combinations, serving as input for these linker design methods. We highlight our main contributions as follows:

- We present a meticulously curated dataset derived from synergistic drug combinations for dual-target drug design, offering new opportunities for AI-driven drug discovery.
- We propose SE(3)-equivariant composed message for compositional generative sampling to reprogram pretrained single-target diffusion models for dual-target drug design in a zero-shot way.
- We propose fragment selection methods from synergistic drug combinations for repurposing linker design methods as strong baselines for dual-target drug design.
- Our method can be viewed as a general framework where any pretrained generative models for SBDD can be applied to dual-target drug design without any fine-tuning. We select TargetDiff as a demonstrative demo in our work.

## 2 Related Work

**Structure-based Drug Design** Structure-based drug design (SBDD) aims to design ligand molecules that can bind to specific protein targets. The introduction of deep generative models has marked a paradigm shift, yielding noteworthy outcomes. Ragoza et al. [48] utilized a variational autoencoder to generate 3D molecules within atomic density grids. Luo et al. [42], Peng et al. [45], Liu et al. [36] employed an autoregressive model to sequentially construct 3D molecules atom by atom, while Zhang et al. [65] introduced a method for generating 3D molecules by successively predicting molecular fragments in an auto-regressive way. Guan et al. [19], Schneuing et al. [53], Lin et al. [33] introduced diffusion models [21] to SBDD, which first generate the types and positions of atoms by iteratively denoising with an SE(3)-equivariant neural network [52, 18] and then determine bond types by post-processing. Some recent studies have endeavored to further improve

the aforementioned methods through the integration of biochemical prior knowledge. Guan et al. [20] proposed decomposed priors, bond diffusion and validity guidance to improve the quality of ligand molecules generated by diffusion models. Zhang and Liu [64] augmented molecule generation through global interaction between subpocket prototypes and molecular motifs. Huang et al. [23] incorporated protein-ligand interaction prior into both forward and reverse processes to improve the diffusion models. Zhou et al. [66] integrated conditional diffusion models with iterative optimization to optimize properties of generated molecules. The above works focus on structure-based single-target drug design, while our work aims at dual-target drug design.

**Molecular Linker Design**   Molecular linker design, which enables the connection of molecular fragments to form potent compounds, is an effective approach in rational drug discovery. Approaches like DeLinker [26] and Develop [27] design linkers by utilizing molecular graphs with distance and angle information between anchor atoms, but they lack 3D structural information of molecules. More recent techniques, such as 3DLinker [22] and DiffLinker [25], generate linkers directly in 3D space using conditional VAEs and diffusion models, respectively, but they assume known fragment poses. LinkerNet [17] relaxes this assumption by co-designing molecular fragment poses and the linker, making it applicable in cases where fragment poses are unknown, such as in the linker design of PROTACs (PROteolysis TArgeting Chimeras). Since pharmacophore combination is a traditional strategy to design dual-target drugs, we repurpose linker design methods as strong baselines for dual-target drug design. Please refer to Appendix B for extended related works.

## 3   Method

In this section, we will present the pipeline of our work, from dataset curation to method. In Section 3.1, we will introduce how we curate the dual-target dataset based on synergistic drug combinations and how we derived the protein-ligand complex structures. In Section 3.2, we will show how we reprogram the pretrained target-specific diffusion models for dual-target drug design and introduce two methods, COMPDIFF and DUALDIFF. In Section 3.3, we will show how we repurpose linker design methods for dual-target drug design.

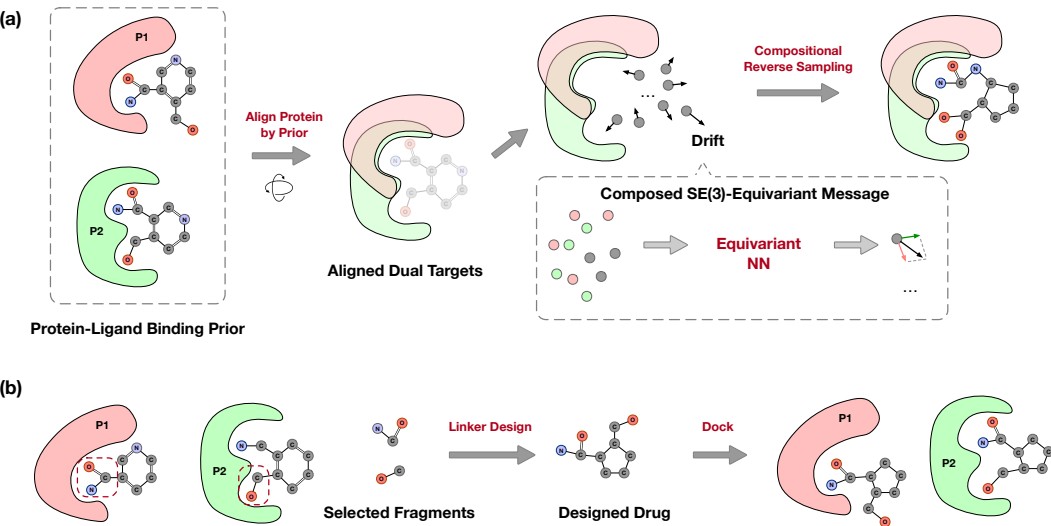

Figure 1: Overview of our method for dual-target drug design. (a) Illustration of COMPDIFF and DUALDIFF. We first align two pockets in 3D space with protein-ligand binding prior and build two complex graph with shared ligand nodes. We then compose the SE(3)-equivariant message to derive the drift on output level (COMPDIFF) or at each layer of the equivariant neural network (DUALDIFF). Based on the composed drift, we can generate dual-target ligand molecules by compositional reverse sampling. (b) Illustration of repurposing linker design methods for dual-target drug design. We first identity binding-related fragments from the reference molecules for each of the dual targets and then apply linker design methods to link the fragments and derive a complete molecule that can bind to the dual targets separately.

### 3.1 Data Curation

Designing dual-target drugs for random pairs of targets lacks significant intent. However, by taking cues from drug synergy, where two drugs together deliver an impact greater than the sum of their separate effects [58], we carefully select target pairs to ensure the dataset holds practical significance for drug discovery.

**Drug Synergy**  To collect drug combination pairs, we start from DrugCombDB[2] [35]. Drug-CombDB is a comprehensive database devoted to the curation of drug combinations from various data sources including high-throughput screening (HTS) assays, manual curations from the literature, FDA Orange Book and external databases. DrugCombDB comprises a total of 448,555 combinations of drugs, encompassing 2,887 unique drugs and 124 human cancer cell lines. Particularly, Drug-CombDB has more than 6,000,000 quantitative dose responses, from which we determine whether a drug combination is synergistic or not. Specifically, a drug combination with positive zero interaction potency (ZIP), Bliss, Loewe and the highest single agent (HSA) scores simultaneously in at least one cell line is supposed to be a synergistic one. Please refer to Appendix C for a comprehensive understanding of these scores.

**Drug Information**  After collecting synergistic drug combinations, we need to collect other necessary information (e.g., SMILES and targets) according to their drug names provided by DrugCombDB. Before this procedure, we collect synonyms and cross-matching ID (e.g., CAS Number and ChEBI ID) mainly from DrugBank[3] [30] and Therapeutic Target Database (TTD)[4] [67]. This step facilitates comprehensive literature reviews, ensuring that all relevant data sources that may use alternate names for a drug is considered. We then collect SMILES or structures (if possible) also mainly from DrugBank and TTD. To identify drug targets, we also utilize DrugBank and TTD as the primary data sources, and supplement these with manual curation from the literature (e.g., [28]). For drugs for which we cannot find either SMILES or targets, we exclude them from our previously collected dataset of positive drug combinations.

**Complex Structures**  For certain drug-target pairs, we incorporate their complex structures directly into our dataset if they are available in PDBBind [38], a repository of protein-ligand binding structures sourced from the Protein Data Bank (PDB) [2]. For drug-target pairs not present in PDBBind, we initially attempt to retrieve the target structures from PDB; if the structures are unavailable, we then source them from the AlphaFold Protein Structure Database (AlphaFold DB) [59] and exclude those whose confidence scores, referred to as pLDDT which provide an assessment of the structures predicted by AlphaFold 2 [29], are less than 70. For these protein targets with structures from PDB or AlphaFold DB, we first utilize P2Rank [31], a program that precisely predicts ligand-binding pockets from a protein structure, to find the most possible pocket given the target structure, and use AutoDock Vina [12] to obtain the protein-ligand complex structures. For each drug, there may exist more than one targets, in which case we use AutoDock Vina to measure the binding affinity and selected the target with the best binding affinity. Finally, we obtain 12,917 postive drug combinations with protein-ligand complex structures, among which there are 438 unique drugs. The 12,917 pairs of targets can be used to evaluate the ability of methods for dual-target drug design. And the binding ligands can be used for reference molecules.

### 3.2 Reprogramming Target-Specific Diffusion Models for Dual-Target Drug Design

Diffusion models [21, 8, 56, 54] have been introduced to structure-based drug design and achieved promising results [19, 53, 33, 20]. We will first revisit the background of diffusion models for SBDD [19] and then introduce how we apply diffusion models trained on single-target protein-ligand datasets to dual-target drug design in a zero-shot manner. Our method is illustrated in Figure 1 (a).

**Diffusion Models for single-target SBDD**  In this following, we denote the type of an atom as $v \in \mathbb{R}^K$ and the coordinate of an atom as $x \in \mathbb{R}^3$, where $K$ is the number of atom types of our interest. For single-target drug design, given a protein binding site denoted as a set of atoms

---

[2]`http://drugcombdb.denglab.org/main`
[3]`https://go.drugbank.com/`
[4]`https://idrblab.net/ttd/`

$\mathcal{P} = \{(\boldsymbol{x}_P^{(i)}, \boldsymbol{v}_P^{(i)})\}_{i=1}^{N_P}$, where $N_P$ is the number of protein atoms, our goal is to generate binding molecules $\mathcal{M} = \{(\boldsymbol{x}_L^{(i)}, \boldsymbol{v}_L^{(i)})\}_{i=1}^{N_M}$. For brevity, we denote the molecule as $M = [\mathbf{x}, \mathbf{v}]$, where $[\cdot, \cdot]$ denotes the concatenation operator and $\mathbf{x} \in \mathbb{R}^{N_M \times 3}, \mathbf{x} \in \mathbb{R}^{N_M \times K}$ denote the coordinates in 3D space and one-hot atom types, respectively. So we can use generative models to model the conditional distribution $p(M|\mathcal{P})$.

In the forward *diffusion* process of the diffusion model, noises are gradually injected into the data sample (i.e., small molecule $M_0 \sim p(M|\mathcal{P})$) and lead to a sequence of latent variable $M_1, M_2, \ldots, M_T$. The final distribution $p(M_T|\mathcal{P})$, also known as prior distribution, is approximately standard normal distribution for atom positions and uniform distribution for atom types. The reverse *generative* process learns to recover data distribution from the noise distribution with a neural network parameterized by $\boldsymbol{\theta}$. The forward and reverse processes are both Markov chains defined as follows:

$$q(M_{1:T}|M_0, \mathcal{P}) = \prod_{t=1}^{T} q(M_t|M_{t-1}, \mathcal{P}) \quad \text{and} \quad p_{\boldsymbol{\theta}}(M_{0:T-1}|M_T, \mathcal{P}) = \prod_{t=1}^{T} p_{\boldsymbol{\theta}}(M_{t-1}|M_t, \mathcal{P}). \quad (1)$$

More specifically, the forward transition kernel in Guan et al. [19] are defined as follows:

$$q(M_t|M_{t-1}, \mathcal{P}) = \mathcal{N}(\mathbf{x}_t; \sqrt{1-\beta_t}\mathbf{x}_{t-1}, \beta_t \boldsymbol{I}) \cdot \mathcal{C}(\mathbf{v}_t|(1-\beta_t)\mathbf{v}_{t-1} + \beta_t/K), \quad (2)$$

where $\{\beta_t\}_{t=1}^{T}$ are fixed noise schedule. The above diffusion process can be efficiently sampled directly from time step 0 to $t$ as follows:

$$q(\mathbf{x}_t|\mathbf{x}_0) = \mathcal{N}(\mathbf{x}_t; \sqrt{\bar{\alpha}_t}\mathbf{x}_0, (1-\bar{\alpha}_t)\boldsymbol{I}) \quad \text{and} \quad q(\mathbf{v}_t|\mathbf{v}_0) = \mathcal{C}(\mathbf{v}_t|\bar{\alpha}_t \mathbf{v}_0 + (1 - \bar{\alpha}_t/K), \quad (3)$$

where $\alpha_t := 1 - \beta_t$ and $\bar{\alpha}_t := \prod_{s=1}^{t} \alpha_s$. The posterior can be easily computed via Bayes theorem as:

$$q(\mathbf{x}_{t-1}|\mathbf{x}_t, \mathbf{x}_0) = \mathcal{N}(\mathbf{x}_{t-1}; \tilde{\boldsymbol{\mu}}_t(\mathbf{x}_t, \mathbf{x}_0), \tilde{\beta}_t \boldsymbol{I}) \quad \text{and} \quad q(\mathbf{v}_t|\mathbf{v}_0) = \mathcal{C}(\mathbf{v}_t|\bar{\alpha}_t \mathbf{v}_0 + (1 - \bar{\alpha}_t)/K), \quad (4)$$

where $\tilde{\beta}_t = \frac{1-\bar{\alpha}_{t-1}}{1-\bar{\alpha}_t}\beta_t$, $\tilde{\mu}_t(\mathbf{x}_t, \mathbf{x}_0) = \frac{\sqrt{\bar{\alpha}_{t-1}}\beta_t}{1-\bar{\alpha}_t}$, $\tilde{c}_t(\mathbf{v}_t, \mathbf{v}_0) = \boldsymbol{c}^* / \sum_{k=1}^{K} c_k^*$ and $\boldsymbol{c}^*(\mathbf{v}_t, \mathbf{v}_0) = [\alpha_t \mathbf{v}_t + (1 - \alpha_t)/K] \odot [\bar{\alpha}_{t-1}\mathbf{v}_0 + (1 - \bar{\alpha}_{t-1})/K]$.

Accordingly, the reverse transition kernel are defined as follows:

$$p_{\boldsymbol{\theta}}(M_{t-1}|M_t, \mathcal{P}) = \mathcal{N}(\mathbf{x}_{t-1}; \boldsymbol{\mu}_{\boldsymbol{\theta}}([\mathbf{x}_t, \mathbf{v}_t], t, \mathcal{P}), \sigma_t^2 \boldsymbol{I}) \cdot \mathcal{C}(\mathbf{v}_{t-1}|\mathbf{c}_{\boldsymbol{\theta}}([\mathbf{x}_t, \mathbf{v}_t], t, \mathcal{P})). \quad (5)$$

Guan et al. [19] use SE(3)-equivariant neural networks [52, 18] to parameterize $\boldsymbol{\mu}_{\boldsymbol{\theta}}([\mathbf{x}_t, \mathbf{v}_t], t, \mathcal{P})$ and $\mathbf{c}_{\boldsymbol{\theta}}([\mathbf{x}_t, \mathbf{v}_t], t, \mathcal{P})$. More specifically, the $[\mathbf{x}_0, \mathbf{v}_0]$ are first predicted using neural network $f_{\boldsymbol{\theta}}$, i.e., $[\hat{\mathbf{x}}_0, \hat{\mathbf{v}}_0] = f_{\boldsymbol{\theta}}([\mathbf{x}_t, \mathbf{v}_t], t, \mathcal{P})$ and then substitute in the posterior as in Equation (4). At the $l$-th layer of $f_{\boldsymbol{\theta}}$, the hidden embedding $\mathbf{h}$ and coordinates $\mathbf{x}$ of each atom are updated alternately as follows:

$$\mathbf{h}_i^{l+1} = \mathbf{h}_i^l + \sum_{j \in \mathcal{V}, i \neq j} f_{\boldsymbol{\theta}_h}(\mathbf{h}_i^l, \mathbf{h}_j^l, d_{ij}^l, \mathbf{e}_{ij}), \quad (6)$$

$$\mathbf{x}_i^{l+1} = \mathbf{x}_i^l + \sum_{j \in \mathcal{V}, i \neq j} (\mathbf{x}_i^l - \mathbf{x}_j^l) f_{\boldsymbol{\theta}_x}(\mathbf{h}_i^{l+1}, \mathbf{h}_j^{l+1}, d_{ij}^l, \mathbf{e}_{ij}) \cdot \mathbf{1}_{\text{ligand}}, \quad (7)$$

where $\mathcal{V}$ is a k-nearest neighbors (knn) graph, $d_{ij} = \|\mathbf{x}_i - \mathbf{x}_j\|$ is the Euclidean distance between two atoms $i$ and $j$, $\mathbf{e}_{ij}$ is an additional feature that indicates the connection is between protein atoms, ligand atoms or protein atom and ligand atom, and $\mathbf{1}_{\text{ligand}}$ is a mask for ligand nodes since only coordinates of ligand atoms are supposed to be updated.

The diffusion model is trained to minimize the KL-divergence between the ground-truth posterior $q(M_{t-1}|M_0, M_t, \mathcal{P})$ and the estimated posterior $p_{\boldsymbol{\theta}}(M_{t-1}|M_t, \mathcal{P})$. After being trained, given a specific pocket, the ligand molecule can be generated by first sampling from prior distribution and sequentially applying the reverse generative process defined above.

**Problem Definition of Dual-Target Drug Design** The goal of dual-target drug design is to design a ligand molecule $M$ that can bind to both given pocket $\mathcal{P}_1$ and pocket $\mathcal{P}_2$. The problem can be also formulated as a generative task which models the conditional distribution $p(M|\mathcal{P}_1, \mathcal{T}\mathcal{P}_2)$. Notably, we introduce a transformation operator $\mathcal{T}$ here. This is because protein pockets exhibit a wide variety of shapes and chemical characteristics and it is necessary to achieve spatial alignment of the dual pockets when modeling the conditional distribution with both of them as conditions. To maintain a neat but siginificant setting, we restrict the transformation $\mathcal{T}$ to encompass solely translations $\mathcal{T}_T$ and rotations $\mathcal{T}_R$, i.e., $\mathcal{T} = \mathcal{T}_T \circ \mathcal{T}_R$. To be more precise, $(\mathcal{T}_T \circ \mathcal{T}_R)\mathcal{P}_2 = \{(\boldsymbol{R}\boldsymbol{x}_{P_2}^{(i)} + \boldsymbol{t}), \boldsymbol{v}_{P_2}^{(i)}\}_{i=1}^{N_{P_2}}$, where $\boldsymbol{R} \in \text{SO}(3)$ represents the rotation and $\boldsymbol{t} \in \mathbb{R}^3$ represents the translation.

**Aligning Dual Targets with Protein-Ligand Binding Priors**   Protein pockets can have intricate 3D structures with varying depths, widths, and surface contours, and distinct chemical properties, including differences in surface electron potentials, hydrophobicity, and the distribution of functional groups. The complex nature of pocket characteristics hinder us from directly aligning two pockets. Nevertheless, the binding mode is determined by the complex nature of pocket information, thus the protein-binding priors can effectively summarize the essential information needed for aligning the two pockets. This motivate us to propose to use a ligand molecule as a prober to implicitly reflect the spatial arrangement of the two pockets and then align them with the protein-ligand binding priors. More specifically, we first dock a ligand molecule to pocket $\mathcal{P}1$ and pocket $\mathcal{P}2$ separately. We can then compute $\boldsymbol{R}$ and $\boldsymbol{t}$ by aligning the two docked poses of the ligand molecule. Experiments have demonstrated that even ligand molecules capable of approximate binding to the two pockets can effectively indicate a specific spatial alignment between them. Further details are provided in Section 4.

**SE(3)-Equivariant Composed Message and Compositional Generative Sampling**   Inspired by compositional visual generation [9, 37], we can model the dual-target drug design with the following composed distribution:

$$p(M|\mathcal{P}_1, \mathcal{T}\mathcal{P}_2) \propto p_{\boldsymbol{\theta}}(M|\mathcal{P}_1)p_{\boldsymbol{\theta}}(M|\mathcal{T}\mathcal{P}_2). \tag{8}$$

Following Liu et al. [37], for the atom position prediction, we can reparameterize $\boldsymbol{\mu}_{\boldsymbol{\theta}}([\mathbf{x}_t, \mathbf{v}_t], t, \mathcal{P})$ with $\mathbf{x}_t - \boldsymbol{\epsilon}_{\boldsymbol{\theta}}([\mathbf{x}_t, \mathbf{v}_t], t, \mathcal{P})$, so that we can rewrite the transition kernel of the reverse generative process as follows:

$$p_{\boldsymbol{\theta}}(\boldsymbol{x}_{t-1}|\boldsymbol{x}_t, \mathcal{P}) = \mathcal{N}(\mathbf{x}_{t-1}; \boldsymbol{\mu}_{\boldsymbol{\theta}}([\mathbf{x}_t, \mathbf{v}_t], t, \mathcal{P}), \sigma_t^2 \boldsymbol{I}) = \mathcal{N}(\mathbf{x}_{t-1}; \mathbf{x}_t - \boldsymbol{\epsilon}_{\boldsymbol{\theta}}([\mathbf{x}_t, \mathbf{v}_t], t, \mathcal{P}), \sigma_t^2 \boldsymbol{I}). \tag{9}$$

The reversed transition kernel corresponds to a step as follow:

$$\mathbf{x}_{t-1} = \mathbf{x}_t - \boldsymbol{\epsilon}_{\boldsymbol{\theta}}([\mathbf{x}_t, \mathbf{v}_t], t, \mathcal{P}) + \mathcal{N}(\mathbf{0}, \sigma_t^2 \boldsymbol{I}). \tag{10}$$

where $\boldsymbol{\epsilon}_{\boldsymbol{\theta}}([\mathbf{x}_t, \mathbf{v}_t], t, \mathcal{P})$ can be viewed as a drift term and $\mathcal{N}(\mathbf{0}, \sigma_t^2 \boldsymbol{I})$ can be viewed as a diffusion term. As Liu et al. [37] points out, this is analogous to the Langevin dynamics [11] that used to sample from Energy-Based Models (EBMs) [10, 55], which can be formulated as follows:

$$\mathbf{x}_{t-1} = \mathbf{x}_t - \frac{\lambda}{2}\nabla_{\mathbf{x}}E_{\boldsymbol{\theta}}([\mathbf{x}_t, \mathbf{v}_t], t, \mathcal{P}) + \mathcal{N}(\mathbf{0}, \sigma_t^2 \boldsymbol{I}). \tag{11}$$

The sampling procedure produces samples from the probability density $p_{\boldsymbol{\theta}}(\mathbf{x}|\mathcal{P}) \propto \exp\left(-E_{\boldsymbol{\theta}}(\mathbf{x}|\mathcal{P})\right)$ where $E_{\boldsymbol{\theta}}(\mathbf{x}|\mathcal{P})$ is a energy function parameterized by model $\boldsymbol{\theta}$. Thus, accordingly, the composed distribution of atom positions can be written as:

$$p(\mathbf{x}|\mathcal{P}_1, \mathcal{T}\mathcal{P}_2) \propto p_{\boldsymbol{\theta}}(\mathbf{x}|\mathcal{P}_1)p_{\boldsymbol{\theta}}(\mathbf{x}|\mathcal{T}\mathcal{P}_2) \propto \exp\left(-\left(E_{\boldsymbol{\theta}}([\mathbf{x}_t, \mathbf{v}_t], t, \mathcal{P}_1) + E_{\boldsymbol{\theta}}([\mathbf{x}_t, \mathbf{v}_t], t, \mathcal{T}\mathcal{P}_2)\right)\right).$$

This composed distribution corresponds to Langevin dynanmics as follows:

$$\mathbf{x}_{t-1} = \mathbf{x}_t - \frac{\lambda}{2}\nabla_{\mathbf{x}}\left(-\left(E_{\boldsymbol{\theta}}([\mathbf{x}_t, \mathbf{v}_t], t, \mathcal{P}_1) + E_{\boldsymbol{\theta}}([\mathbf{x}_t, \mathbf{v}_t], t, \mathcal{T}\mathcal{P}_2)\right)\right). \tag{12}$$

Accordingly, each step in the compositional reverse generative sampling process can be defined as:

$$\mathbf{x}_{t-1} = \mathbf{x}_t - \eta\left(\left(\boldsymbol{\epsilon}_{\boldsymbol{\theta}}([\mathbf{x}_t, \mathbf{v}_t], t, \mathcal{P}_1) + \boldsymbol{\epsilon}_{\boldsymbol{\theta}}([\mathbf{x}_t, \mathbf{v}_t], t, \mathcal{P}_2)\right)\right), \tag{13}$$

where we additionally introduce a hyperparameter $\eta$ here to control the strength of the drift. In theory, this is equivalent to making a more flexible assumption that $p(\mathbf{x}|\mathcal{P}_1, \mathcal{T}\mathcal{P}_2) \propto [p_{\boldsymbol{\theta}}(\mathbf{x}|\mathcal{P}_1)p_{\boldsymbol{\theta}}(\mathbf{x}|\mathcal{T}\mathcal{P}_2)]^{\eta}$. In this case the transition kernel is defined as $p_{\boldsymbol{\theta}}(\mathbf{x}_{t-1}|\mathbf{x}_t, \mathcal{P}_1, \mathcal{T}\mathcal{P}_2) = [p_{\boldsymbol{\theta}}(\mathbf{x}_{t-1}|\mathbf{x}_t, \mathcal{P}_1)p_{\boldsymbol{\theta}}(\mathbf{x}_{t-1}|\mathbf{x}_t, \mathcal{T}\mathcal{P}_2)]^{\eta}$. In practice, we set $\eta = 1/2$ by default. This composition operation is equivalent to averaging two $\hat{\mathbf{x}}_0$ predicted on two complex graphs, i.e. $\mathcal{V}_1$ and $\mathcal{V}_2$. Similarly, for the atom types which are discrete variables, we can also compose the transition kernel as follows: $p_{\boldsymbol{\theta}}(\mathbf{v}_{t-1}|\mathbf{v}_t, \mathcal{P}_1, \mathcal{T}\mathcal{P}_2) \propto p_{\boldsymbol{\theta}}(\mathbf{v}_{t-1}|\mathbf{v}_t, \mathcal{P}_1)p_{\boldsymbol{\theta}}(\mathbf{v}_{t-1}|\mathbf{v}_t, \mathcal{T}\mathcal{P}_2)$. Note that $p_{\boldsymbol{\theta}}(\mathbf{v}_{t-1}|\mathbf{v}_t, \mathcal{P}_1)$ is categorical distribution and its dimension (i.e., the number of atom types of our interest) is $K$, which is small in practice. So $p_{\boldsymbol{\theta}}(\mathbf{v}_{t-1}|\mathbf{v}_t, \mathcal{P}_1, \mathcal{T}\mathcal{P}_2)$ can be computed analytically. We name the compositional reverse sampling with composed transition kernel (i.e., composed drift) as COMPDIFF.

We further improve the compositional reverse sampling by introducing the composition into each layer of the equivariant neural network in the pretrained diffusion model. For brevity, we denote the SE(3)-equivariant message at the $l$-th layer for the $i$-th atom of the complex graph $\mathcal{V}_v$ ($v = 1, 2$) as introduced in Equation (7) as follows:

$$\Delta \mathbf{h}_i^l(\mathcal{V}_v) := \sum_{j \in \mathcal{V}_v, i \neq j} f_{\boldsymbol{\theta}_h}(\mathbf{h}_i^l, \mathbf{h}_j^l, d_{ij}^l, \mathbf{e}_{ij}), \tag{14}$$

$$\Delta \mathbf{x}_i^l(\mathcal{V}_v) := \sum_{j \in \mathcal{V}_v, i \neq j} (\mathbf{x}_i^l - \mathbf{x}_j^l) f_{\boldsymbol{\theta}_x}(\mathbf{h}_i^{l+1}, \mathbf{h}_j^{l+1}, d_{ij}^l, \mathbf{e}_{ij}) \cdot \mathbf{1}_{\text{ligand}}. \tag{15}$$

The above SE(3)-equivariant message can also be interpreted as drift in 3D and latent space. Thus we can also compose them as follows:

$$\mathbf{h}_i^{l+1} = \mathbf{h}_i^l + \left(\Delta \mathbf{h}_i^l(\mathcal{V}_1) + \Delta \mathbf{h}_i^l(\mathcal{V}_2)\right)/2 \quad \text{and} \quad \mathbf{x}_i^{l+1} = \mathbf{x}_i^l + \left(\Delta \mathbf{x}_i^l(\mathcal{V}_1) + \Delta \mathbf{x}_i^l(\mathcal{V}_2)\right)/2. \tag{16}$$

We name the compositional reverse sampling with the above SE(3)-equivariant message at each layer as DUALDIFF. The proof of SE(3)-equivariance can be found in Appendix G. This more meticulous composition is supposed to lead to higher-quality samples.

### 3.3 Repurposing Linker Design Methods for Dual-Target Drug Design

Pharmacophore combination is a prevalent strategy in traditional dual-target drug design, requiring the specialized knowledge of chemists. To automate this procedure, We design a strategy to identity crucial fragments from reference molecules of dual targets in our dataset and apply linker design methods [25, 17] to link the fragments and obtain complete molecules. Please refer to Appendix D for a more detailed justification for the choice of baselines.

Specifically, we break all rotatable bonds of reference molecule $\mathcal{M}_1$ (resp. $\mathcal{M}_2$) of target $\mathcal{P}_1$ (resp. $\mathcal{P}_2$) to obtain fragments. Since DiffLinker [25] requires relative positions of fragments as input, we dock all fragments derived from $\mathcal{M}_1$ and $\mathcal{M}_2$ to $\mathcal{P}_2$ (or $\mathcal{P}_1$) and select the pair of fragments which has the best sum of binding affinity and no physical conflicts (i.e., the minimum between atoms from the two fragments is large than 1.4Å). We then apply DiffLinker to link the two fragments, considering the existence of the pocket $\mathcal{P}_2$ (or $\mathcal{P}_1$). Since LinkerNet [17] models the translation and rotation of fragments by neural networks and does not require relative position of fragments as input, we directly dock all fragments derived from $\mathcal{M}_1$ (resp. $\mathcal{M}_2$) to $\mathcal{P}_1$ (resp. $\mathcal{P}_2$) and select their respective fragment with the best binding affinity. And we then link them using LinkerNet to obtain complete molecules.

## 4 Experiment

### 4.1 Experimental Setup

**Dataset** We use our dataset introduced in Section 3.1. All 12,917 pairs of targets (including 438 unique targets) are used for evaluation. For each target, there is an associated reference molecule that can be considered a benchmark for high-quality ligand molecules and utilized in linker design methods.

**Baselines** We compare our method with various baselines: **Pocket2Mol** [45] generates 3D molecules atom by atom in an autoregressive manner given a specific protein binding site. **TargetDiff** [19] is a diffusion-based method which generates atom coordinates and atom types in a non-autoregressive way. Note that Pocket2Mol and TargetDiff are both proposed for structure-based single-target drug design. **DiffLinker** [25] is a diffusion-based model for linker design with given fragment poses. **LinkerNet** [17] is a diffusion-based model for co-designing molecular fragment poses and the linker. DiffLinker and LinkerNet are repurposed for dual-target drug design as we introduced in Section 3.3. The code is available at `https://github.com/zhouxiangxin1998/DualDiff`.

**Evaluation** We evaluate generated ligand molecules from the perspectives of target binding affinity and molecular properties. We employ AutoDock Vina [12] to estimate the target binding affinity, following Peng et al. [45], Guan et al. [19]. We first evaluate Pocket2Mol and TargetDiff under the

Table 1: Summary of different properties of reference molecules and molecules generated by baselines and our methods under the **dual-target** setting. (↑) / (↓) denotes a larger / smaller number is better. Top 2 results are highlighted with **bold text** and underlined text, respectively.

| Methods | P-1 Vina Dock (↓) Avg. | P-1 Vina Dock (↓) Med. | P-2 Vina Dock (↓) Avg. | P-2 Vina Dock (↓) Med. | Max Vina Dock (↓) Avg. | Max Vina Dock (↓) Med. | Dual High Aff. (↑) Avg. | Dual High Aff. (↑) Med. | QED (↑) Avg. | QED (↑) Med. | SA (↑) Avg. | SA (↑) Med. | Diversity (↑) Avg. | Diversity (↑) Med. |
|---|---|---|---|---|---|---|---|---|---|---|---|---|---|---|
| Reference | -7.60 | -7.80 | -6.02 | -7.30 | -5.46 | -7.09 | - | - | 0.53 | 0.54 | 0.74 | 0.77 | - | - |
| Pocket2Mol | -4.82 | -4.76 | -4.63 | -4.64 | -4.40 | -4.42 | 0.2% | 0.0% | 0.49 | 0.48 | **0.88** | **0.90** | **0.82** | **0.83** |
| TargetDiff | **-8.62** | **-8.61** | -6.89 | -7.67 | -6.57 | -7.39 | 29.0% | 20.0% | 0.50 | 0.51 | 0.58 | 0.58 | 0.70 | 0.71 |
| DiffLinker | -7.05 | -7.87 | -7.27 | -7.92 | -5.87 | -7.18 | 24.6% | 0.0% | 0.43 | 0.42 | 0.30 | 0.29 | 0.52 | 0.54 |
| LinkerNet | -8.20 | -8.37 | -8.13 | -8.38 | -7.17 | -7.72 | 35.7% | 0.0% | **0.61** | **0.63** | 0.72 | 0.73 | 0.37 | 0.34 |
| COMPDIFF | -8.32 | -8.42 | -8.37 | -8.47 | -7.50 | -7.78 | 35.9% | 30.0% | 0.55 | 0.57 | 0.59 | 0.59 | 0.72 | 0.72 |
| DUALDIFF | -8.41 | -8.51 | **-8.48** | **-8.55** | **-7.66** | **-7.88** | **36.3%** | **30.2%** | 0.56 | 0.58 | 0.59 | 0.59 | 0.67 | 0.67 |

single-target setting as a preliminary verification (see Appendix E). For dual-target drug design, we use each method to design 10 molecules for each pair of targets, denoted as $\mathcal{P}_1$ and $\mathcal{P}_2$. (For reference molecule, Pocet2Mol and TargetDiff, the ligand molecule is generated for $\mathcal{P}_1$ but the target binding affinity is evaluated on both $\mathcal{P}_1$ and $\mathcal{P}_2$.) We then collect all generated molecules across 12,917 pairs of targets and report the mean and median (denoted as "Avg." and "Med." respectively) of affinity-related metrics (P-1 Vina Dock, P-2 Vina Dock, Max Vina Dock, and Dual High Affinity) and property-related metrics (drug-likeness QED [3], synthesizability SA [14], and diversity). Vina Dock incorporates a re-docking step to assess the highest binding affinity achievable. Here we introduce P-1 Vina Dock and P-2 Vina Dock to represent the Vina Dock score evaluated on $\mathcal{P}_1$ and $\mathcal{P}_2$, respectively. Besides, we introduce Max Vina Dock, which represents the maximum Vina Dock of a given molecule towards $\mathcal{P}_1$ and $\mathcal{P}_2$. The Vina Dock will be low if and only if the molecule can bind to both targets simultaneously, which is the goal of dual-target drug design. Additionally, we report Dual High Affinity (abbreviated as Dual High Aff.) which represents the proportion of generated molecules that exhibit binding affinity that exceeds that of the reference molecules on both the respective targets. This reflects the success rate in achieving higher binding affinities simultaneously on both targets in dual-target drug design. We also evaluate the RMSD between docked poses towards dual targets.

## 4.2 Main Results

We compare all methods under the dual-target setting. The results are reported in Table 1. Our methods, COMPDIFF and DUALDIFF, significantly outperforms other methods, especially in terms of binding affinity. Notably, DUALDIFF achieves the highest Dual High Affinity among all methods. In line with our expectations, for the single-target drug design methods (e.g., Pocket2Mol and TargetDiff), we observe a significant decline in performance according to P-2 Vina Dock compared to P-1 Vina Dock, which shows their inability in dual-target drug design. LinkerNet also achieves promising results except diversity. Note that DiffLinker and LinkerNet are provided with reference molecules while COMPDIFF and DUALDIFF are not. This indicates the strong generative abilities of our methods. Finally, DUALDIFF outperforms COMPDIFF, which shows that composition of SE(3)-equivariant message at each layer is more effective than only at the output level.

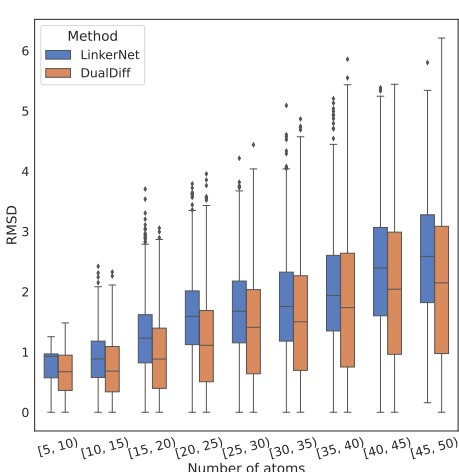

Figure 2: RMSD between docked poses towards dual targets of different methods.

As shown in Figure 2, DUALDIFF performs better than LinkerNet the RMSD between docked poses towards dual targets. This indicates that the molecules generated by DUALDIFF can bind to dual targets with smaller conformation change. We also benchmark the inference time of baslines and our methods in Appendix H. Additionally, we provide visualization of examples of generated molecules in Figure 3. See more examples in Appendix F.

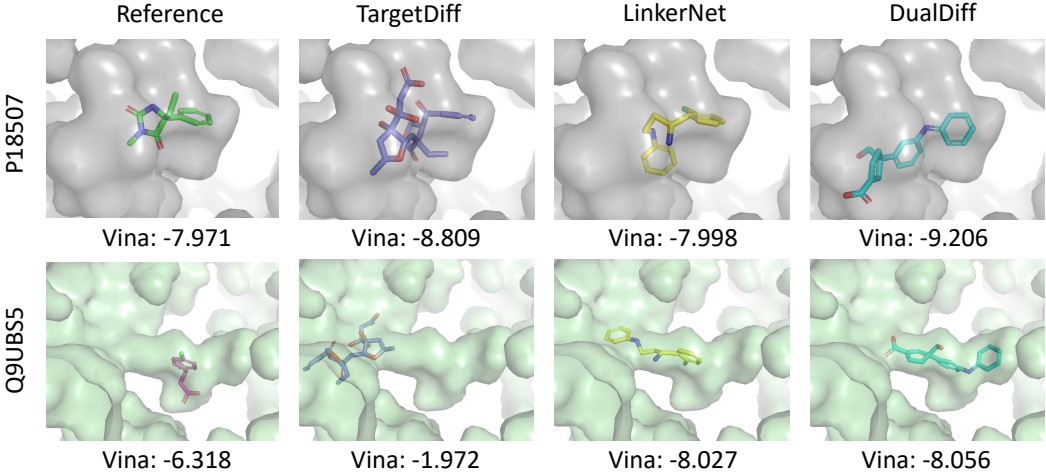

Figure 3: Reference molecules and examples of ligand molecules by different methods generated for the dual targets (UniProt ID: P18507 (top) and Q9UBS5 (bottom)).

Table 2: Ablation on different ways of aligning dual targets.

| Methods | P-1 Vina Dock (↓) | | P-2 Vina Dock (↓) | | Max Vina Dock (↓) | | Dual High Aff. (↑) | | QED (↑) | | SA (↑) | | Diversity (↑) | |
|---|---|---|---|---|---|---|---|---|---|---|---|---|---|---|
| | Avg. | Med. | Avg. | Med. | Avg. | Med. | Avg. | Med. | Avg. | Med. | Avg. | Med. | Avg. | Med. |
| COMPDIFF-Center | -8.10 | -8.26 | -8.08 | -8.26 | -7.23 | -7.60 | 30.8% | 22.0% | 0.52 | 0.53 | 0.60 | 0.59 | 0.73 | 0.73 |
| COMPDIFF-RMSD | -8.29 | -8.44 | -8.35 | -8.46 | -7.47 | -7.78 | 35.6% | 30.0% | 0.55 | 0.56 | 0.59 | 0.59 | 0.72 | 0.72 |
| COMPDIFF-Score | -8.32 | -8.42 | -8.37 | -8.47 | -7.50 | -7.78 | 35.9% | 30.0% | 0.55 | 0.57 | 0.59 | 0.59 | 0.72 | 0.72 |
| DUALDIFF-Center | -8.12 | -8.29 | -8.12 | -8.28 | -7.32 | -7.66 | 30.0% | 22.2% | 0.52 | 0.54 | 0.59 | 0.59 | 0.69 | 0.69 |
| DUALDIFF-RMSD | -8.40 | -8.51 | -8.45 | -8.53 | -7.63 | -7.87 | 35.8% | 30.0% | 0.55 | 0.57 | 0.59 | 0.59 | 0.67 | 0.67 |
| DUALDIFF-Score | -8.41 | -8.51 | -8.48 | -8.55 | -7.66 | -7.88 | 36.3% | 30.2% | 0.56 | 0.58 | 0.59 | 0.59 | 0.67 | 0.67 |

Table 3: Ablation on different strategies of identifying fragments for linker design methods.

| Methods | P-1 Vina Dock (↓) | | P-2 Vina Dock (↓) | | Max Vina Dock (↓) | | Dual High Aff. (↑) | | QED (↑) | | SA (↑) | | Diversity (↑) | |
|---|---|---|---|---|---|---|---|---|---|---|---|---|---|---|
| | Avg. | Med. | Avg. | Med. | Avg. | Med. | Avg. | Med. | Avg. | Med. | Avg. | Med. | Avg. | Med. |
| DiffLinker-5 | -6.74 | -6.74 | -6.85 | -6.81 | -6.21 | -6.32 | 9.4% | 0.0% | 0.58 | 0.60 | 0.32 | 0.31 | 0.57 | 0.62 |
| DiffLinker-pocket-5 | -7.35 | -7.74 | -7.20 | -7.75 | -6.25 | -7.14 | 20.2% | 0.0% | 0.45 | 0.45 | 0.31 | 0.30 | 0.58 | 0.62 |
| DiffLinker-8 | -7.22 | -7.32 | -7.49 | -7.49 | -6.65 | -6.86 | 15.7% | 0.0% | 0.61 | 0.63 | 0.32 | 0.32 | 0.49 | 0.51 |
| DiffLinker-pocket-8 | -7.05 | -7.87 | -7.27 | -7.92 | -5.87 | -7.18 | 24.6% | 0.0% | 0.43 | 0.42 | 0.30 | 0.29 | 0.52 | 0.54 |
| LinkerNet-5 | -7.54 | -7.54 | -7.56 | -7.58 | -6.98 | -7.05 | 19.9% | 0.0% | 0.69 | 0.72 | 0.77 | 0.79 | 0.46 | 0.46 |
| LinkerNet-self-5 | -7.54 | -7.55 | -7.55 | -7.56 | -6.98 | -7.05 | 20.0% | 0.0% | 0.69 | 0.72 | 0.77 | 0.79 | 0.46 | 0.46 |
| LinkerNet-8 | -8.09 | -8.23 | -8.23 | -8.31 | -7.29 | -7.63 | 34.0% | 0.0% | 0.62 | 0.65 | 0.72 | 0.74 | 0.38 | 0.35 |
| LinkerNet-self-8 | -8.20 | -8.37 | -8.13 | -8.38 | -7.17 | -7.72 | 35.7% | 0.0% | 0.61 | 0.63 | 0.72 | 0.73 | 0.37 | 0.34 |

## 4.3 Ablation Studies

**Effects of Alignment of Dual Targets** We perform different methods to align the pockets of dual targets for COMPDIFF and DUALDIFF. See the results in Table 2. Naively, we can align two pockets by their geometric centers (denoted as "-Center"). In our method, we propose to align pockets with protein-ligand binding priors. We select the ligand with minimum RMSD between docked poses (resp. minimum sum of Vina Dock scores) towards dual targets as the anchor to align the dual targets, which is denoted as "-RMSD" (resp. "-Score"). DUALDIFF-Score achieved the best performance among all variants, demonstrating the effectiveness of the alignment method.

**Different Strategies of Identifying Fragments for Linker Design Methods** We conduct ablation on different strategies of identifying fragments for DiffLinker and LinkerNet for dual-target drug design. Since we use Vina Dock to select fragments, we have tried different box sizes for docking, i.e., 5Å and 8Å. For DiffLinker, since the relative poses of fragments are required as input, we dock fragments derived from $\mathcal{M}_1$ and $\mathcal{M}_2$ to target $\mathcal{P}_1$ (or $\mathcal{P}_2$). We then apply DiffLinker both with and without considering the pocket. The corresponding methods are denoted as "DiffLinker-5/-8/-pocket-5/-pocket-8". For LinkerNet, the relative poses of fragments are not required. So we try two setting: dock fragments from $\mathcal{M}_1$ and $\mathcal{M}_2$ to target $\mathcal{P}_1$ (or $\mathcal{P}_2$); dock fragments from $\mathcal{M}_1$ (resp. $\mathcal{M}_2$) to target $\mathcal{P}_1$ (resp. $\mathcal{P}_2$). The difference is that all fragments are docked to the same pocket in the former setting while the fragments from two ligand molecules are docked to their respective pockets. The

corresponding methods are denoted as "LinkerNet-5/-8/-self-5/-self-8". The results are reported in Table 3. The results show that a larger docking box size allows for better selection of fragments. As we expected, DiffLinker with consideration of pockets achieves better performance. Among all variants, "LinkerNet-self-8" achieves best performance, which implies that adjusting relative poses of fragments may play a crutial role in linker deisgn. This feature of LinkerNet allows for selecting the most important fragments for each pockets of the dual targets.

## 5    Conclusion

In this work, we introduced a novel dataset for dual-target drug design. We formulate this problem as a generative task and propose compositional reverse sampling to reprogram pretrained target-specific model for dual-target drug design, which can successfully generate dual-target ligands and outperform all baselines, including repurposed linker design methods. Our research lays the groundwork for dual-target drug design using generative methods, with future progress expected.

## Acknowledgments

This work is supported by the National Science and Technology Major Project (2023ZD0120901) and the National Natural Science Foundation of China No. 62377030. We also thank the reviewers for their valuable feedback.

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

# A  Motivation, Significance, and Current Practices of Dual-Target Drug Design

**Motivation and significance of dual-target drug design**  The reviewers and audience can refer to Giordano and Petrelli [16], Boran and Iyengar [5] and [41] for the concept of dual-target (or multi-target) drugs. Dual-target drugs have various advantages, especially in aspects such as anti-tumor and overcoming drug resistance.

It has been revealed that single-target drugs may not always induce the desired effect on the entire biological system even if they successfully inhibit or activate a specific target [47]. One reason is that organisms can affect effectiveness through compensatory ways. Another reason is the development of resistance either by self-modification of the target through mutation or by the adoption of new pathways by a cancer cell, for the growth and multiplication [47]. Cancer and those nervous and cardiovascular system diseases are complicated, thereby promoting the multi-targeted therapies as a better pathway to achieve the desired treatment. For example, the multiple tyrosine kinase inhibitor imatinib[5] induces better anti-cancer effects compared with that of gefitinib[6], which involves a single target, further indicating that drugs with multiple targets may exhibit a better chance of affecting the complex equilibrium of whole cellular networks than drugs that act on a single target. Refer to Lu et al. [41] for more examples. Notably, combination therapy also performs well in overcoming drug resistance [15]. However, under some circumstances, the combination therapy might have additive and even synergistic effects in theory; however, it often leads to some unpredictable side effects, such as increased toxicity. Thus, multi-target drugs are a better choice.

**Current practices for dual-target drug design**  Refer to Sun et al. [57] for a comprehensive review. The mainstream strategies are drug repurposing, skeleton modification, and linked/merged/fused pharmacophores, which involve complicated pipelines and require domain knowledge from experts. To the best of our knowledge, we are the first one to formulate dual-target drug design as a generative task and provide simple yet effective solutions.

# B  Extended Related Works

Our work is also relevant to neural model reprogramming[7]. We will discuss the connections as follows:

Elsayed et al. [13] proposed adversarial reprogramming. Specifically, given a model trained on a task which maps $x$ to $f(x)$, the adversary aims to repurpose the model for a new task which maps $g(\tilde{x})$ for inputs $\hat{x}$, by learning mapping functions $h_f(\cdot; \theta)$ and $h_g(\cdot; \theta)$ to make $h_g(f(h_f(\hat{x})))$ approximate $g(\hat{x})$. They successfully demonstrated their methods on image classification tasks. For example, a trained adversarial program can cause a classifier trained on ImageNet to an MNIST classifier.

Yang et al. [61] reprograms acoustic models for time series classification, through input transformation learning and output label mapping, which are similar to $h_f(\cdot)$ and $h_g(\cdot)$, respectively. The motivation is that data scarcity hinders researchers from using large-scale deep learning models for time-series tasks while many large-scale pre-trained speech processing models are available. Melnyk et al. [43] also follows this paradigm to reprogram pretrained language models, BERT trained on English corpus, for the antibody sequence infilling task.

The above works all aim to reprogram a model trained on a task to perform a new task. Our work also falls into this paradigm, since we focus on reprogramming models trained for single-target drug design to design dual-target drugs. And we also have a similar motivation of Yang et al. [61] and Melnyk et al. [43], because there is no training data for dual-target drug design. Differently, we focus on reprogramming a diffusion model equipped with SE(3)-equivariance which has not been widely explored in previous works due to its complex sampling process during inference. And, notably, all the above works require additional training to learn the additional parameters of input transformation (i.e., $h_f(\cdot; \theta)$ in Elsayed et al. [13]) for reprogramming. However, our framework works in a zero-shot manner, which means we only need to modify the sampling process of the diffusion model to perform

---

[5]https://en.wikipedia.org/wiki/Imatinib

[6]https://en.wikipedia.org/wiki/Gefitinib

[7]Note that the reprogramming here is not relevant to the biological concept "cell reprogramming" [6].

the novel task without any additional training. Specifically, our proposed pocket alignment resembles the "input transformation", i.e., $h_f(\cdot)$, and the compositional sampling resembles the "output label mapping", i.e., $h_g(\cdot)$, but no learnable parameters are introduced for reprogramming. Yang et al. [61] provided theoretical results on selecting a pre-trained model for reprogramming, but it is not easy to directly apply the results to our case due to the intricate properties of diffusion models.

## C Term Definitions in Drug Synergy

**Drug Synergy**   Drug synergy refers to the phenomenon where two or more drugs used in combination, produce a more therapeutic effect than the sum of their individual effects. When drugs with distinct binding targets are strategically paired, they can leverage each other's strengths and compensate for respective weaknesses, which enables lower individual drug doses, thereby reducing the risk of adverse effects. This characteristic is usually used in cancer and HIV treatments, and has been proven to be a new but promising way to combat complex diseases [7].

**Zero Interaction Potency (ZIP)**   Yadav et al. [60] proposed ZIP score to describe the drug interaction by comparing the alteration in the potency of dose-response curves in the context of single drug administration versus their concurrent use in combinations. In the two-drug combination scenario, we will refer to drug A and drug B, respectively. The effects of these drugs are defined as $E_{AB}$ for combination, $E_A$ and $E_B$ for individual situations ($0 \leq E \leq 1$). The ZIP score can be defined as follows:

$$S_{\text{ZIP}} = \bar{E}_{\text{AB}} - (\bar{E}_{\text{A}} + \bar{E}_{\text{B}} - \bar{E}_{\text{A}} \cdot \bar{E}_{\text{B}}). \tag{17}$$

The $\bar{E}_{\text{AB}}$ in Equation (17) is the average response values obtained by fitting dose-response curves independently in each dimension of the measured combinatorial data sub-tensor, the explanation for the other variables are the same.

**Bliss**   The Bliss independence model [24] assumes a stochastic process where the two drugs elicit their effects independently. In this model, the expected combination effect can be calculated based on the probability of the independent events occurring:

$$S_{\text{Bliss}} = E_{AB} - (1 - (1 - E_A)(1 - E_B)). \tag{18}$$

Equation (18) is similar to Equation (17), the difference between which is that Equation (17) employs fitted drug responses instead of observed ones.

**Loewe**   The Loewe score [51] forecasts the dose combination that will produce a specific effect, it calculates the expected response as if both drugs are the same. Assume drug A can produce effect $E_A$ at dose $x_A$, and drug B can produce effect $E_B$ at dose $x_B$, then the loewe affinity states that the expected effect $E_{\text{Loewe}}$ can be determined by:

$$\frac{x_A}{X_A} + \frac{x_B}{X_B} = 1, \tag{19}$$

where $X_A$ and $X_B$ are the doses drug A or B alone that produces effect $E_{\text{Loewe}}$. The Loewe score is then defined as follows:

$$S_{\text{Loewe}} = E_{\text{AB}} - E_{\text{Loewe}}. \tag{20}$$

**Highest Single Agent (HSA)**   The HSA is a straightforward scoring system for estimating drug synergy, which determines the incremental effect of combining drugs by comparing the enhanced combined effect to their individual effects [60]. The HSA score can be calculated as follows:

$$S_{\text{HSA}} = E_{\text{AB}} - \max(E_A, E_B) \tag{21}$$

The ZIP score is the most prevalently utilized metric in the assessment of drug synergy. Furthermore, these scores can be collectively analyzed to identify optimal drug combinations for targeted therapy.

## D Justification for the Choice of Baselines

One of the traditional strategies for dual-target drug design (refer to Sun et al. [57] as a comprehensive review) is linking pharmacophores by domain experts, such as chemists or pharmaceutical scientists.

We use SOTA deep-learning-based linker design methods (DiffLinker [25] and LinkerNet [19]) to mimic this procedure. Since each target pocket has a corresponding ligand molecule in our curated dataset, we can break the ligand molecule into fragments and select the ones critical for binding as pharmacophores. In this case, given respective critical fragments of dual pockets, we can design linkers and resemble them into complete molecules. Ideally, the designed molecules contain critical pharmacophores that account for binding to dual targets and serve as a potential candidate for dual-target drugs.

## E    Preliminary Verification on Single-Target Setting

We evaluate Pocket2Mol and TargetDiff under the setting of single-target drug design on 438 unique targets in our dataset as a preliminary verification. We use each method generates 10 molecules for each target and collect all generated molecules across 438 proteins and report the mean, trimmed mean (i.e., averaging that removes 10% of the largest and smallest values before calculating the mean) and median (denoted as "Avg.", "T-Avg." and "Med." respectively) of affinity-related metrics (Vina Score, Vina Min, Vina Dock, and High Affinity) and property-related metrics (drug-likeness QED [3], synthesizability SA [14], and diversity). Vina Score assesses binding affinity directly based on the generated 3D molecules. Vina Min carries out a energy minimization over the local structure before estimation. Vina Dock incorporates a re-docking step to assess the highest binding affinity achievable. Meanwhile, High Affinity evaluates the proportion of generated molecules that have stronger binding affinity than the reference molecule for each protein tested.

The results are shown in Table 4, where molecules generated by TargetDiff exhibits binding affinities that are either comparable to or slightly greater than those of the reference molecules. Pocket2Mol achieves strong performance in terms of QED, SA, and diversity but fails in Vina-related metrics. Therefore, TargetDiff can be regarded as an effective molecular generative model on our dataset.

Table 4: Summary of different properties of reference molecules and molecules generated by baselines under the **single-target** setting. ($\uparrow$) / ($\downarrow$) denotes a larger / smaller number is better. Considering outliers significantly affect the mean, we ignore some abnormal values of Vina Score and Vina Min.

| Methods | Vina Score ($\downarrow$) | | | Vina Min ($\downarrow$) | | | Vina Dock ($\downarrow$) | | | High Affinity ($\uparrow$) | | QED ($\uparrow$) | | SA ($\uparrow$) | | Diversity ($\uparrow$) | |
|---|---|---|---|---|---|---|---|---|---|---|---|---|---|---|---|---|---|
| | Avg. | T-Avg. | Med. | Avg. | T-Avg. | Med. | Avg. | T-Avg. | Med. | Avg. | Med. | Avg. | Med. | Avg. | Med. | Avg. | Med. |
| Reference | - | -8.02 | -7.96 | - | -8.07 | -8.04 | -8.09 | -8.37 | -8.25 | - | - | 0.54 | 0.55 | 0.74 | 0.78 | - | - |
| Pocket2Mol | -3.14 | -3.19 | -3.16 | -3.81 | -3.77 | -3.76 | -4.87 | -4.86 | -4.85 | 2.1% | 0.0% | 0.49 | 0.49 | 0.87 | 0.89 | 0.83 | 0.85 |
| TargetDiff | - | -7.38 | -7.38 | - | -7.89 | -7.86 | -8.77 | -8.78 | -8.75 | 56.0% | 55.6% | 0.51 | 0.53 | 0.58 | 0.58 | 0.69 | 0.70 |

## F    Visualization of More Examples

Here we provide visualization of more examples as shown in Figure 4.

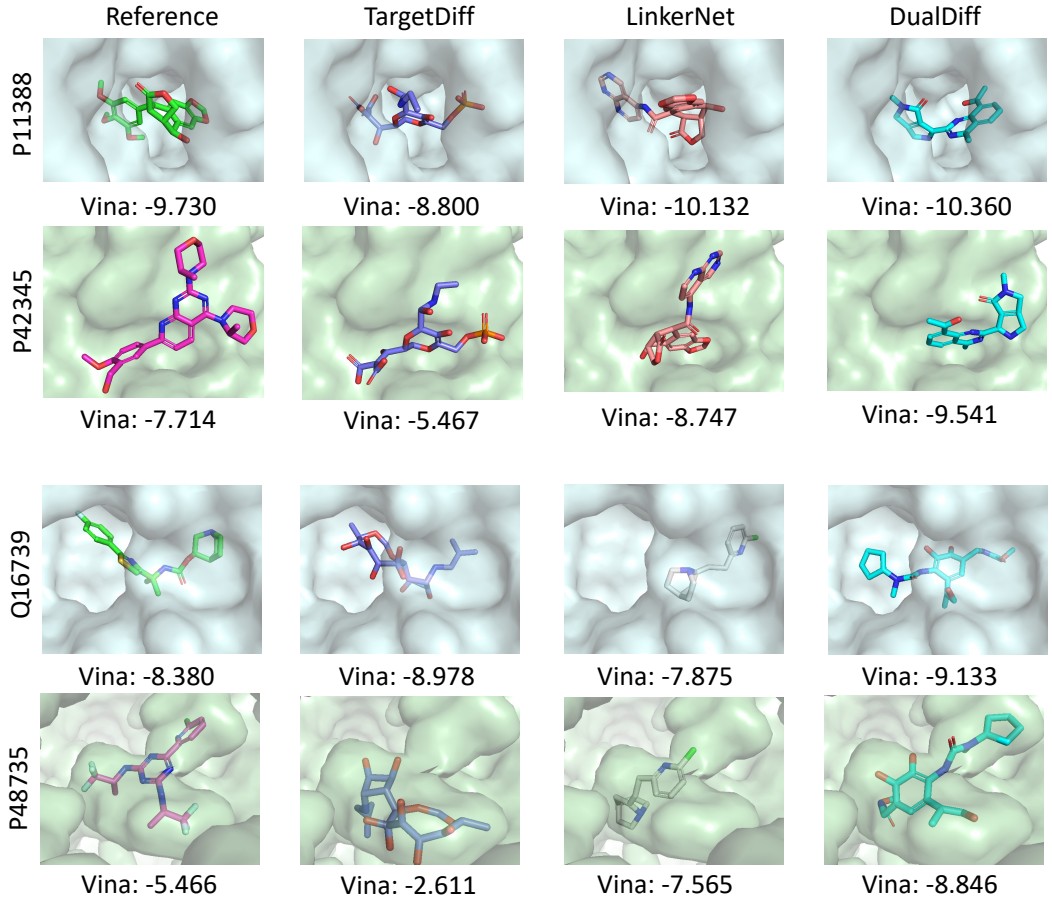

Figure 4: Visualization of more reference molecules and examples designed by TargetDiff, LinkerNet and DUALDIFF.

# G   Proof of SE(3)-Equivariance

We denote the global SE(3) transformation as $T_g$, and which means the transformation as $T_g(\mathbf{x}_i) = \boldsymbol{R}_g \mathbf{x}_i + \boldsymbol{b}$, where $\boldsymbol{R}_g \in \mathbb{R}^{3\times 3}$ is the rotation matrix and $\boldsymbol{b} \in \mathbb{R}^3$ is the translation vector. In line with Section 3.2, we define the composed message passing as follows:

$$\Delta \mathbf{h}_i^l(\mathcal{V}_v) := \sum_{j\in\mathcal{V}_v, i\neq j} f_{\boldsymbol{\theta}_h}(\mathbf{h}_i^l, \mathbf{h}_j^l, d_{ij}^l, \mathbf{e}_{ij}) \tag{22}$$

$$\Delta \mathbf{x}_i^l(\mathcal{V}_v) := \sum_{j\in\mathcal{V}_v, i\neq j} (\mathbf{x}_i^l - \mathbf{x}_j^l) f_{\boldsymbol{\theta}_x}(\mathbf{h}_i^{l+1}, \mathbf{h}_j^{l+1}, d_{ij}^l, \mathbf{e}_{ij}) \cdot \mathbf{1}_{\text{ligand}} \tag{23}$$

$$\mathbf{h}_i^{l+1} = \mathbf{h}_i^l + \frac{1}{2}\big(\Delta \mathbf{h}_i^l(\mathcal{V}_1) + \Delta \mathbf{h}_i^l(\mathcal{V}_2)\big) \tag{24}$$

$$\mathbf{x}_i^{l+1} := \phi(\mathbf{x}_i^l) = \mathbf{x}_i^l + \frac{1}{2}(\Delta \mathbf{x}_i^l(\mathcal{V}_1) + \Delta \mathbf{x}_i^l(\mathcal{V}_2)) \tag{25}$$

It is easy to see that the atomic distance $d_{ij}^l = \|\mathbf{x}_i - \mathbf{x}_j\|$ and $\mathbf{e}_{ij}$ feature are both invariant to SE(3) transformation The hidden embedding $\mathbf{h}_i^l$ is also invariant since the its updates (as shown in Equation (22) and Equation (24)) are only related to invariant features.

We define $\Delta T_g\big(\mathbf{x}_i^l(\mathcal{V}_v)\big)$ as

$$
\begin{aligned}
\Delta T_g\big(\mathbf{x}_i^l(\mathcal{V}_v)\big) &:= \sum_{j\in\mathcal{V}_v, i\neq j} \big(T_g(\mathbf{x}_i^l) - T_g(\mathbf{x}_i^l)\big) f_{\boldsymbol{\theta}_x}(\mathbf{h}_i^{l+1}, \mathbf{h}_j^{l+1}, d_{ij}^l, \mathbf{e}_{ij}) \cdot \mathbf{1}_{\text{ligand}} \\
&= \sum_{j\in\mathcal{V}_v, i\neq j} \big(\boldsymbol{R}_g\mathbf{x}_i^l + \boldsymbol{b} - \boldsymbol{R}_g\mathbf{x}_j^l - \boldsymbol{b}\big) f_{\boldsymbol{\theta}_x}(\mathbf{h}_i^{l+1}, \mathbf{h}_j^{l+1}, d_{ij}^l, \mathbf{e}_{ij}) \cdot \mathbf{1}_{\text{ligand}} \\
&= \sum_{j\in\mathcal{V}_v, i\neq j} \boldsymbol{R}_g\big(\mathbf{x}_i^l - \mathbf{x}_j^l\big) f_{\boldsymbol{\theta}_x}(\mathbf{h}_i^{l+1}, \mathbf{h}_j^{l+1}, d_{ij}^l, \mathbf{e}_{ij}) \cdot \mathbf{1}_{\text{ligand}}
\end{aligned}
$$

After applying $T_g$ to $\mathbf{x}_i^l$, the updated position $\mathbf{x}_i^{l+1} = \phi(\mathbf{x}_i^l)$ can be written as (using the above results):

$$
\begin{aligned}
\phi\big(T_g(\mathbf{x}_i^l)\big) &= T_g(\mathbf{x}_i^l) + \frac{1}{2}\big(\Delta T_g(\mathbf{x}_i^l(\mathcal{V}_1)) + \Delta T_g(\mathbf{x}_i^l(\mathcal{V}_2))\big) \\
&= \boldsymbol{R}_g\mathbf{x}_i^l + \boldsymbol{b} + \frac{1}{2}\Bigg( \sum_{v\in 1,2} \sum_{j\in\mathcal{V}_v, i\neq j} \boldsymbol{R}_g\big(\mathbf{x}_i^l - \mathbf{x}_j^l\big) f_{\boldsymbol{\theta}_x}(\mathbf{h}_i^{l+1}, \mathbf{h}_j^{l+1}, d_{ij}^l, \mathbf{e}_{ij}) \cdot \mathbf{1}_{\text{ligand}} \Bigg) \\
&= \boldsymbol{R}_g\mathbf{x}_i^l + \frac{1}{2}\boldsymbol{R}_g\Bigg( \sum_{v\in 1,2} \sum_{j\in\mathcal{V}_v, i\neq j} \big(\mathbf{x}_i^l - \mathbf{x}_j^l\big) f_{\boldsymbol{\theta}_x}(\mathbf{h}_i^{l+1}, \mathbf{h}_j^{l+1}, d_{ij}^l, \mathbf{e}_{ij}) \cdot \mathbf{1}_{\text{ligand}} \Bigg) + \boldsymbol{b} \\
&= \boldsymbol{R}_g\mathbf{x}_i^l + \frac{1}{2}\boldsymbol{R}_g\big(\Delta\mathbf{x}_i^l(\mathcal{V}_1) + \Delta\mathbf{x}_i^l(\mathcal{V}_2)\big) + \boldsymbol{b} \\
&= \boldsymbol{R}_g\bigg(\mathbf{x}_i^l + \frac{1}{2}\big(\Delta\mathbf{x}_i^l(\mathcal{V}_1) + \Delta\mathbf{x}_i^l(\mathcal{V}_2)\big)\bigg) + \boldsymbol{b} \\
&= T_g\bigg(\mathbf{x}_i^l + \frac{1}{2}\big(\Delta\mathbf{x}_i^l(\mathcal{V}_1) + \Delta\mathbf{x}_i^l(\mathcal{V}_2)\big)\bigg) \\
&= T_g\big(\phi(\mathbf{x}_i^l)\big)
\end{aligned}
$$

The above equation shows the SE(3)-equivariance of the atom position update formula Equation (25). Based on the fact that Equation (24) is SE(3)-invariant and Equation (25) is SE(3)-equivariant, we can say that the composition operation of our method is SE(3)-equivariant.

## H   Computational Efficiency

We test the time for generating 10 molecules by TargetDiff (single-target drug design) and DiffLinker, LinkerNet, CompDiff, and DualDiff (dual-target drug design). The results are shown in Table 5.

Table 5: Time Cost for generating 10 molecules by TargetDiff (single-target drug design) and DiffLinker, LinkerNet, CompDiff, and DualDiff (dual-target drug design).

| Setting | Method | Time (seconds) |
|---|---|---|
| Single-Target | TargetDiff | 220 |
| Dual-Target | DiffLinker | 31 |
|  | LinkerNet | 30 |
|  | COMPDIFF | 395 |
|  | DUALDIFF | 493 |

Linker design methods (DiffLInker and LinkerNet) are faster than de novo design methods. The reasons are twofold: 1. For linker design, parts of the molecules are provided; thus, it is only necessary to generate the remaining part (i.e., the linker). 2. For DiffLinker, only one of the dual targets can serve as the input of the model. For linkerNet, it cannot use pockets as conditions. Thus, the computational cost is lower.

As the table shows, the inference time of CompDiff and DualDiff (dual-target drug design) is about twice as that of TargetDiff (single-target drug design). This is as expected because there are two heterogeneous graphs and the messages on the two graphs are processed and aggregated separately

(sequentially in our implementation) and then composed as described in our paper. There is space for optimizing the inference speed of CompDiff and DualDiff by parallelling the message-passing operations over the two graphs (when GPU memory is sufficient). This code optimization will only speed up the inference and not change the generated results. Ideally, this will make the inference time of CompDiff and DualDiff almost the same as that of TargetDiff.

Though the computational cost of our method is higher than the baselines, it is still acceptable in practice.

General acceleration methods (e.g., pruning, quantization, and distillation) for deep learning models can be applied to the models used in our framework. Besides, fast diffusion sampling solvers can also be applied to our framework, with DPM-Solver [40] as one of the most representative works, which can significantly reduce the number of sampling steps without sacrificing the quality of generated samples.

# I   Discussion, Limitation, and Future Work

Our work provides a novel dataset and a general framework for dual-target drug design to the community. And our method can be easily adapted to the multi-target scenario. Our work is the first step towards generative dual-target drug design. There are still limitations in our work. For example, we do not consider flexibility of proteins in our work, which is a more practical setting, though this is also an issue for most works in SBDD. Another limitation of our work is the lack of validation through wet-lab experiments. We will leave these as future work.

# J   Societal Impacts

Our research holds the promise of significantly advancing the pharmaceutical industry by aiding in the development of potent dual-target drugs. This could potentially streamline the path to new treatments, making efficient drug discovery a more attainable goal. Moreover, emphasizing the ethical implementation of our methods is of paramount importance. It is also needed to ensure that these scientific achievements are utilized for social good, safeguarding against any misuse that could lead to negative consequences for society.

