# OpenReview forum: "Reprogramming Pretrained Target-Specific Diffusion Models for Dual-Target Drug Design"
_NeurIPS.cc/2024/Conference — NeurIPS 2024 poster_

### Official Review · Reviewer_7yyK · 2024-06-17

**Soundness:** 3
**Presentation:** 3
**Contribution:** 2
**Rating:** 5
**Confidence:** 4

**Summary:**

This is an interesting and promising work that combines state-of-the-art techniques in structure-based drug design, such as diffusion models, to address the challenging but important problem of dual-target drug design. The main contributions of the paper are:

1.Carefully curated a dataset based on synergistic drug combinations for the dual-target drug design task, offering new opportunities for AI-driven drug discovery.
2. Proposed SE(3)-equivariant composed message passing to reprogram pretrained single-target diffusion models for dual-target drug generation in a zero-shot manner.
3. Proposed fragment selection methods from synergistic drug combinations to repurpose linker design as baselines for dual-target drug design.
4. The method can be viewed as a general framework where generative models for single-target SBDD can be applied to dual-target design without fine-tuning.

**Strengths:**

1. The paper addresses an important and challenging problem in drug discovery, namely dual-target drug design, which has the potential to overcome drug resistance and manage complex diseases more effectively.
2. The authors curate a novel dataset based on synergistic drug combinations, providing a valuable resource for AI-driven dual-target drug discovery.
3. The proposed method, which reprograms pretrained single-target diffusion models for dual-target drug design using SE(3)-equivariant composed message passing, is innovative and has the potential to transfer knowledge from single-target to dual-target scenarios without the need for extensive training data.
4. The authors repurpose linker design methods as strong baselines for the dual-target drug design task and propose strategies to identify potential fragments from synergistic drug combinations, demonstrating a comprehensive approach to the problem.

**Weaknesses:**

1. The method description is unclear in several aspects:
a) The formal definition of the dual-target drug design problem, especially the pocket alignment, is lacking.
b) The specific approach for aligning two pockets is not well explained, leaving questions about the consideration of different ligands and the quality of the alignment.
c) The composed reverse process in the generative model is not described in sufficient detail, making it difficult to understand how it is defined and sampled.
2. The experimental results are not presented in the paper, making it impossible to assess the effectiveness of the proposed method. The authors should include detailed quantitative and qualitative results and compare their method with baseline approaches to demonstrate its advantages.

**Questions:**

1. What are the current practices and needs for dual-target drug design in industry? The authors could discuss the importance of this task in drug discovery, as well as the limitations and future directions of this work.

**Limitations:**

1. The motivation and significance of dual-target design need to be further emphasized. The current discussion in the Introduction about the advantages of dual-target drugs is not sufficiently convincing. More references should be added to support the advantages of dual-target drugs, especially in aspects such as anti-tumor and overcoming drug resistance.

2. The novelty of the proposed generative models need to be further enhanced. Compared to the existing approaches, the proposed methods has no inspiring contributions except SE(3)-equivariant modeling, which is also a common idea in this field.

---

> ### Author Rebuttal · Authors · 2024-08-07
>
> Thank you for your feedback. Please see below for our responses to the comments.
>
> **Q1: Formal definition of the dual-target drug design problem, especially the pocket alignment, and consideration of different ligands and the quality of the alignment.**
>
> A1: We have provided the definition of the problem in line 184-192. Here we discuss it in more detail:
>
> For single-target drug design, it can be formulated as a generative task. Specifically, following Pocket2Mol and TargetDiff, the goal is to model the conditional distribution $P(M|\mathcal{P})$, where $\mathcal{P}$ is a protein pocket with its 3D structure and $M$ is the ligand molecule with its 3D structure. Autoregressive models (e.g., Pocket2Mol) and diffusion models (e.g., TargetDiff) are developed for the 3D molecule generation task.
>
> Intuitively, the dual-target drug design can also be formulated as a generative task and the goal is to model the conditional distribution $P(M|\mathcal{P_1}, \mathcal{P_2})$, where $\mathcal{P}_1, \mathcal{P}_2$ are the two different pockets. However, this formula is ill-defined when considering the 3D structures, because $\mathcal{P}_1$ and $\mathcal{P}_2$ do not lie in the same coordinate system and any geometric operations are meaningless. Thus, we need to introduce a transformation $\mathcal{T}$ to align the two pockets so that they lie in the same coordinate. In this case, the goal is to model $P(M|\mathcal{P}_1, \mathcal{T}\mathcal{P}_2)$.
>
> Considering that single-target model is trained on protein-ligand complex datasets, to alleviate the gap between training (single-target) and inference (dual-target), we propose to align the two pockets by the protein-ligand interaction prior. Specifically, this alignment ensures the ligand molecule being generated can stay in a spatial position that is suitable for both pockets. We have described how to obtain such $\mathcal{T}$ in practice in line 201-205. Here we provide a more mathematical description: We first select a molecule $M_1$ that binds to pocket $\mathcal{P}\_1$ and then we dock it to pocket $\mathcal{P}\_2$ to obtain $M_2$. Note that $M_1, M_2$ are the same molecule with different 3D poses. The transformation can be defined as  $\mathcal{T}:= \text{argmin}_{\mathcal{T}} \text{RMSD}(M_1, \mathcal{T}M_2)$.
>
> Then we provided ablation studies on how we choose the anchor molecule $M_1$ for obtaining the transformation. See Section 4.2 and Table 2 for details. The results show that our proposed alignment methods can ensure the model generates valid molecules and consistently surpass the baseline.
>
> **Q2: "The composed reverse process in the generative model is not described in sufficient detail, making it difficult to understand how it is defined and sampled."**
>
> A2: We have described the composed reverse process in detail in line 206-239 of the paper.
>
> In the paper, we describe the process from the perspective of energy-based (or score-based) generative models and Langevin dynamics, which are the motivation of the compositional operation. The description explains why the compositional operation at each step in the process can generate samples from the distribution $P(M|\mathcal{P}_1)\cdot P(M|\mathcal{T}\mathcal{P}_2)$ in theory.
>
> Though the theoretical justification seems complicated, the practical implementation of our method is actually simple yet effective. In the following, we provided a simplified description from the perspective of diffusion models to help the audience less familiar with diffusion models (or score-based generative models) better understand the operation.
>
> The step at time $t$ of the reverse process is as follows (refer to Equation (4) in the paper for definition of notations):
> $$q(\mathbf{x}\_{t-1}|\mathbf{x}\_{t},\hat{\mathbf{x}}\_0)=\mathcal{N}(\mathbf{x}\_{t-1};\hat{{\mu}}\_t(\mathbf{x}\_t,\hat{\mathbf{x}}\_0),\tilde{\beta}\_t\mathbf{I})$$
> $$q(\mathbf{v}\_{t-1}|\mathbf{v}\_t,\hat{\mathbf{v}}\_0)=\mathcal{C}(\tilde{{c}}\_t(\mathbf{v}\_t,\mathbf{v}\_0))$$
>
> where $\hat{\mathbf{x}}\_0, \hat{\mathbf{v}}\_0$ are the atom positions and types predicted by the SE(3)-equivariant neural network based on $\mathbf{x}\_{t},\mathbf{v}\_{t}$. In the dual-target setting, there are two graphs, based on which two sets of $\hat{\mathbf{x}}\_0, \hat{\mathbf{v}}\_0$ are predicted. The compositional operation composes the two sets of prediction to obtain a composed version of $\hat{\mathbf{x}}\_0, \hat{\mathbf{v}}\_0$. CompDiff directly composes the final outputs of the neural network. DualDiff composes them at each layer of the neural network. By analysis in Section 3.2, the composition is equivalent to 'average' operation over drifts in Euclidean space for atom positions and logorithms of categorical distribution for atom types, which can ensure correct sampling from the target distribution.

---

> ### Author Response · Authors · 2024-08-07
> **Response to Reviewer 7yyK (2/4)**
>
> **Q3: "The experimental results are not presented in the paper, making it impossible to assess the effectiveness of the proposed method. The authors should include detailed quantitative and qualitative results and compare their method with baseline approaches to demonstrate its advantages."**
>
> A3: We have provided both detailed quantitative and qualitative results of the baselines and our methods. See our response to reviewer em5U's comments (Q4 & A4).
>
> See Table 3 for the main results, where we report evaluated molecular properties, including P-1 (resp. P-2) Vina score, max Vina score (over two targets), dual high affinity, QED, SA, and diversity. The entries named 'DiffLinker' and 'LinkerNet' correspond to the repurposed linker design methods for dual-target drug design. The entries named 'CompDiff' and 'DualDiff' correspond to our methods.
>
> See Figure 2 for evaluation from the perspective of molecular conformation.
>
> See Table 2 for ablation studies on different pocket alignment methods.
>
> See Table 3 for ablation studies on different hyperparameters and settings of repurposed linker design methods.
>
> The experimental results show that our methods can generate molecules that bind to dual targets and are consistently superior to baselines.
>
> We also provided examples of generated dual-target drugs designed by various methods in Figures 3 and 4.
>
> Note that the baselines require reference fragments as input while our methods do not, which is also an obvious advantage in cases where no reference fragment or molecule is available.

---

> ### Author Response · Authors · 2024-08-07
> **Response to Reviewer 7yyK (3/4)**
>
> **Q4: Motivation, significance, and current practices of dual-target drug design.**
>
> A4: **Motivation and significance of dual-target drug design:** The reviewers and audience can refer to [1,2,3] for the concept of dual-target (or multi-target) drugs. Dual-target drugs have various advantages, especially in aspects such as anti-tumor and overcoming drug resistance.
>
> It has been revealed that single-target drugs may not always induce the desired effect on the entire biological system even if they successfully inhibit or activate a specific target [4]. One reason is that organisms can affect effectiveness through compensatory ways. Another reason is the development of resistance either by self-modification of the target through mutation or by the adoption of new pathways by a cancer cell, for the growth and multiplication [5]. Cancer and those nervous and cardiovascular system diseases are complicated, thereby promoting the multi-targeted therapies as a better pathway to achieve the desired treatment. For example, the multiple tyrosine kinase inhibitor imatinib [6] induces better anti-cancer effects compared with that of gefitinib [7], which involves a single target, further indicating that drugs with multiple targets may exhibit a better chance of affecting the complex equilibrium of whole cellular networks than drugs that act on a single target. Refer to [3] for more examples. Notably, combination therapy also performs well in overcoming drug resistance [8]. However, under some circumstances, the combination therapy might have additive and even synergistic effects in theory; however, it often leads to some unpredictable side effects, such as increased toxicity. Thus, multi-target drugs are a better choice.
>
> **Current practices for dual-target drug design:** Refer to [9] for a comprehensive review. The mainstream strategies are drug repurposing, skeleton modification, and linked/merged/fused pharmacophores, which involve complicated pipelines and require domain knowledge from experts. To the best of our knowledge, we are the first one to formulate dual-target drug design as a generative task and provide simple yet effective solutions.
>
> **References:**
>
> [1] Giordano, Silvia, and Alessio Petrelli. "From single-to multi-target drugs in cancer therapy: when aspecificity becomes an advantage." Current medicinal chemistry 15, no. 5 (2008): 422-432.
>
> [2] Boran, Aislyn DW, and Ravi Iyengar. "Systems approaches to polypharmacology and drug discovery." Current opinion in drug discovery & development 13, no. 3 (2010): 297.
>
> [3] Lu, Jin-Jian, Wei Pan, Yuan-Jia Hu, and Yi-Tao Wang. "Multi-target drugs: the trend of drug research and development." PloS one 7, no. 6 (2012): e40262.
>
> [4] Puls, Lauren N., Matthew Eadens, and Wells Messersmith. "Current status of SRC inhibitors in solid tumor malignancies." The oncologist 16, no. 5 (2011): 566-578.
>
> [5] Raghavendra, Nulgumnalli Manjunathaiah, Divya Pingili, Sundeep Kadasi, Akhila Mettu, and S. V. U. M. Prasad. "Dual or multi-targeting inhibitors: The next generation anticancer agents." European journal of medicinal chemistry 143 (2018): 1277-1300.
>
> [6] https://en.wikipedia.org/wiki/Imatinib
>
> [7] https://en.wikipedia.org/wiki/Gefitinib
>
> [8] Fischbach, Michael A. "Combination therapies for combating antimicrobial resistance." Current opinion in microbiology 14, no. 5 (2011): 519-523.
>
> [9] Sun, Dejuan, Yuqian Zhao, Shouyue Zhang, Lan Zhang, Bo Liu, and Liang Ouyang. "Dual-target kinase drug design: Current strategies and future directions in cancer therapy." European Journal of Medicinal Chemistry 188 (2020): 112025.
>
> **Q5: Limitations and future directions of this work.**
>
> A5: We have discussed the limitations and future directions in Appendix E and mentioned this in the attached NeurIPS paper checklist (see 2. Limitations).

---

> ### Author Response · Authors · 2024-08-07
> **Response to Reviewer 7yyK (4/4)**
>
> **Q6: "The novelty of the proposed generative models need to be further enhanced. Compared to the existing approaches, the proposed methods has no inspiring contributions except SE(3)-equivariant modeling, which is also a common idea in this field."**
>
> A6: To avoid misunderstandings, we need to emphasize that we do not propose new generative models but a **novel sampling algorithm** including the **pocket alignment** and **compositional operation**. Our proposed method can effectively reprogram single-target diffusion models for dual-target drug design **in a zero-shot manner**, which means that we can design ligand molecules that bind to two targets separately using single-target model without any additional training. This is the contribution of our method instead of SE(3)-equivariant modeling. SE(3)-equivariance is indeed a common choice in this field. We mention it in our paper to show that our proposed method preserves the nice property. And we also provide the corresponding mathematical proof.
>
> In addition to the proposed algorithms, our contribution also lies in the **curated dataset**, which is derived from synergistic drug combinations for dual-target drug design, a task with practical significance and substantial impact, **offering new opportunities for AI-driven drug discovery and computational biology.** This problem has not been widely explored and **we are the first to address this problem by deep learning methods as a generative task. More advanced methods can be developed based on our curated dataset and proposed framework. We expect that our paper will have a significant impact after publication.**

---

> > ### Author Response · Authors · 2024-08-13
> > **Gentle Reminder**
> >
> > Thank you again for your hard work in reviewing our paper!
> >
> > We hope you've had a chance to review our responses to your comments. Please let us know if you have any further questions or concerns. We greatly appreciate your feedback and are committed to addressing any potential issues.

---

> > ### Comment · Reviewer_7yyK · 2024-08-13
> >
> > Thank you very much for your replies to these question. Through your valuable replies, I have realized your results and motivations behind. The problem setting and corresponding data construction will be helpful for the society. However, the math induction and related sampling algorithm, in my personal view, cannot convince me as a "solid and novel" improvement. Thus, to summarize, I decide to raise my score to 5.

---

> ### Author Response · Authors · 2024-08-13
>
> Thank you for your positive support!
>
> Our math induction and the sampling algorithm are inspired by [1], a reference that we have cited in our paper many times. The theoretical analysis is based on Langevin dynamics, as shown in our paper (see line 206-239 for details).
>
> To provide you with a brief understanding, our algorithm can sample from $p_\theta(M|\mathcal{P_1}) \cdot p_\theta(M|\mathcal{T}\mathcal{P_2})$ with a trained diffusion model that directly models $p_\theta(M|\mathcal{P_1})$ and $p_\theta(M|\mathcal{T}\mathcal{P_2})$, which is non-trivial. And we share a similar theoretical justification with [1] which focuses on compositional visual generation.
>
> Differently, our case is more challenging because we need to align the pockets rationally and extend the framework to discrete variables (e.g. atom types) in addition to continuous variables (e.g., atom positions). These challenges come from geometric deep learning and the well-crafted SE(3)-equivariance which visual generation does not have. To our best knowledge, we are the first to introduce the concept of "compositional generation" to the field of AI for drug design or even AI for science. Based on the above analysis, our algorithm is solid and novel.
>
> Thus, we sincerely hope you can reconsider the evaluation of our submission.
>
>
> References:
>
> [1] Liu, Nan, Shuang Li, Yilun Du, Antonio Torralba, and Joshua B. Tenenbaum. "Compositional visual generation with composable diffusion models." In European Conference on Computer Vision, pp. 423-439. Cham: Springer Nature Switzerland, 2022.

---

### Official Review · Reviewer_em5U · 2024-07-13

**Soundness:** 3
**Presentation:** 2
**Contribution:** 3
**Rating:** 4
**Confidence:** 2

**Summary:**

This paper addresses the challenge of designing dual-target drugs, a promising strategy in overcoming drug resistance in cancer therapy. The authors propose leveraging the success of deep generative models in structure-based drug design, formulating dual-target drug design as a generative task. They introduce a novel dataset of potential target pairs derived from synergistic drug combinations and propose a method using diffusion models trained on single-target protein-ligand complex pairs. Their algorithm effectively transfers knowledge from single-target pretraining to dual-target scenarios in a zero-shot manner. Multiple experiments show the effectiveness of their approach compared to various baselines.

**Strengths:**

The formulation of dual-target drug design as a generative task using diffusion models is novel and addresses a significant challenge in cancer therapy. The curation of a dataset with potential target pairs based on synergistic drug combinations provides a valuable resource for the research community. The use of SE(3)-equivariant composed message passing to align two pockets in 3D space and build complex graphs is a strong point, leveraging advanced geometric deep learning techniques. The ability to transfer knowledge from single-target pretraining to dual-target scenarios in a zero-shot manner is impressive, showcasing the model’s flexibility and generalization capability.
The extensive experimental validation against various baselines highlights the robustness and effectiveness of the proposed method.

**Weaknesses:**

The proposed method involves complex geometric deep learning techniques, which may pose implementation challenges for researchers less familiar with these methods.
The computational demands for training and inference using diffusion models and SE(3)-equivariant message passing could be significant, potentially limiting the method’s accessibility.

**Questions:**

How does the computational efficiency of your method compare with the baselines? Are there specific optimizations that can be applied to reduce computational demands?
Can you provide more detailed comparisons and analyses of the repurposed linker design methods used as baselines? How do they perform relative to your proposed method?

**Limitations:**

The paper mentions repurposing linker design methods as strong baselines but does not provide detailed comparisons or analyses of these methods.

The scalability of the proposed method to larger datasets and more complex protein-ligand interactions needs to be explored further.

---

> ### Author Rebuttal · Authors · 2024-08-07
>
> Thank you for your detailed feedback. Please see below for our responses to the comments.
>
> **Q1: "The proposed method involves complex geometric deep learning techniques, which may pose implementation challenges for researchers less familiar with these methods."**
>
> A1:
> To understand our method, researchers need to have a background in both geometric deep learning [1] and diffusion models [2]. To help researchers who are less familiar with these concepts use our method in practice, we will open-source both the code, curated dataset, and generated molecules upon acceptance of this paper and provide a user-friendly interface.
>
> **References:**
>
> [1] Bronstein, Michael M., Joan Bruna, Taco Cohen, and Petar Veličković. "Geometric deep learning: Grids, groups, graphs, geodesics, and gauges." arXiv preprint arXiv:2104.13478 (2021).
>
> [2] Yang, Ling, Zhilong Zhang, Yang Song, Shenda Hong, Runsheng Xu, Yue Zhao, Wentao Zhang, Bin Cui, and Ming-Hsuan Yang. "Diffusion models: A comprehensive survey of methods and applications." ACM Computing Surveys 56, no. 4 (2023): 1-39.
>
> **Q2: "The computational demands for training and inference using diffusion models and SE(3)-equivariant message passing could be significant, potentially limiting the method’s accessibility."**
>
> A2:
> Our framework allows for transferring the knowledge gained in single-target pretraining to dual-target scenarios in a zero-shot manner. More specifically, the open-sourced diffusion model checkpoints (e.g., DiffSBDD [1], TargetDiff [2]) can be directly used in our framework for inference without any additional training or fine-tuning, which can save the computation demanded for training. Actually, the computational demand for training a diffusion model for single-target drug design is acceptable. For example, the training of TargetDiff converges within 24 hours on one NVIDIA GeForce GTX 3090 GPU.
>
> **References:**
>
> [1] Schneuing, Arne, Yuanqi Du, Charles Harris, Arian Jamasb, Ilia Igashov, Weitao Du, Tom Blundell et al. "Structure-based drug design with equivariant diffusion models." arXiv preprint (2022).
>
> [2] Guan, Jiaqi, Wesley Wei Qian, Xingang Peng, Yufeng Su, Jian Peng, and Jianzhu Ma. "3D Equivariant Diffusion for Target-Aware Molecule Generation and Affinity Prediction." ICLR (2023).
>
> **Q3: "How does the computational efficiency of your method compare with the baselines? Are there specific optimizations that can be applied to reduce computational demands?"**
>
> A3: We test the time for generating 10 molecules by TargetDiff (single-target drug design) and DiffLinker, LinkerNet, CompDiff, and DualDiff (dual-target drug design). The results are shown as follows:
>
> | setting | method | Time (seconds) |
> |---|---|---|
> | single-target | TargetDiff | 220 |
> | dual-target | DiffLinker | 31 |
> |  | LinkerNet | 30 |
> |  | CompDiff | 395 |
> |  | DualDiff | 493 |
>
> Linker design methods (DiffLInker and LinkerNet) are faster than de novo design methods. The reasons are twofold: 1. For linker design, parts of the molecules are provided; thus, it is only necessary to generate the remaining part (i.e., the linker). 2. For DiffLinker, only one of the dual targets can serve as the input of the model. For linkerNet, it cannot use pockets as conditions. Thus, the computational cost is lower.
>
> As the table shows, the inference time of CompDiff and DualDiff (dual-target drug design) is about twice as that of TargetDiff (single-target drug design). This is as expected because there are two heterogeneous graphs and the messages on the two graphs are processed and aggregated separately (sequentially in our implementation) and then composed as described in our paper. **There is space for optimizing the inference speed of CompDiff and DualDiff by parallelling the message-passing operations over the two graphs (when GPU memory is sufficient).** This code optimization will only speed up the inference and not change the generated results. Ideally, this will make the inference time of CompDiff and DualDiff almost the same as that of TargetDiff.
>
> Though the computational cost of our method is higher than the baselines, it is still acceptable in practice.
>
> General acceleration methods (e.g., pruning, quantization, and distillation) for deep learning models can be applied to the models used in our framework. Besides, fast diffusion sampling solvers can also be applied to our framework, with DPM-Solver [1] as one of the most representative works, which can significantly reduce the number of sampling steps without sacrificing the quality of generated samples.
>
> **References:**
>
>  [1] Lu, Cheng, Yuhao Zhou, Fan Bao, Jianfei Chen, Chongxuan Li, and Jun Zhu. "Dpm-solver: A fast ode solver for diffusion probabilistic model sampling in around 10 steps." Advances in Neural Information Processing Systems 35 (2022): 5775-5787.

---

> ### Author Response · Authors · 2024-08-07
> **Response to Reviewer em5U (2/2)**
>
> **Q4: Detailed comparisons and analyses of the repurposed linker design methods used as baselines.**
>
> A4: We have provided both quantitative and qualitative results of the repurposed linker design methods. See Table 1 where we report P-1 (resp. P-2) Vina scores, Max Vina scores, dual high affinity, QED, SA, and diversity of all methods including the baselines. The entries named 'DiffLinker' and 'LInkerNet' are the corresponding results of repurposed linker design methods used as baselines. The most important metric is dual high affinity, which represents the ratio of generated molecules that surpass the reference molecules in terms of binding affinity for both protein targets, respectively. We highlight the results as follows:
>
> | Methods | Max Vina Dock ($\downarrow$)  |  | Dual High Affinity ($\uparrow$) |  |
> |---|---|---|---|---|
> |  | Avg. | Med. | Avg. | Med. |
> | reference | -5.46 | -7.09 | - | - |
> | DiffLinker | -5.87 | -7.18 | 24.6% | 0.0% |
> | LInkerNet | -7.17 | -7.72 | 35.7% | 0.0% |
> | CompDiff | -7.50 | -7.78 | 35.9% | 30.0% |
> | DualDiff | -7.66 | -7.88 | 36.3% | 30.2% |
>
> As the results show, our methods are superior to the repurposed linker design methods in dual-target drug design. Notably, we have tried different hyperparameters and settings for the repurposed linker design methods (see Table 3) and report the best configuration in the main experimental results (see Table 1).
>
> In addition to the **molecular properties**, we have also analyzed and compared these methods  from the perspective of **molecular conformation**. Specifically, we calculate RMSD between docked poses towards dual targets of different methods (see Figure 2). Our methods are consistently better than the repurposed linker design methods.
>
> We have also shown specific cases of designed molecules by different methods in Figures 3 and 4, which also show the superiority of our methods.
>
> **Q5: "The scalability of the proposed method to larger datasets and more complex protein-ligand interactions needs to be explored further."**
>
> A5: The scalability is mainly related to the inference speed which we have discussed in A3 & Q3.
> We have discussed the limitations in Appendix E, where we have mentioned that we do not consider the flexibility of proteins and would leave it as future work, which allows for considering more complex protein-ligand interactions, such as induce-fit effects [1].
>
> **Reference:**
>
> [1] Sherman, Woody, Tyler Day, Matthew P. Jacobson, Richard A. Friesner, and Ramy Farid. "Novel procedure for modeling ligand/receptor induced fit effects." Journal of medicinal chemistry 49, no. 2 (2006): 534-553.

---

> > ### Author Response · Authors · 2024-08-13
> > **Gentle Reminder**
> >
> > Thank you for the time and effort you have put into evaluating our submission!
> >
> > We kindly remind you that the discussion phase is coming to the end. Please let us know if there are additional concerns we can address for you. If we have properly addressed your concerns, we sincerely hope you can reconsider the evaluation of our submission.

---

> > > ### Author Response · Authors · 2024-08-13
> > > **Gentle Reminder**
> > >
> > > Dear Reviewer em5U,
> > >
> > > Thank you again for providing valuable comments on our paper!
> > >
> > > Since **the discussion period will come to the end in less than 20 hours**, we hope you've had a chance to review our responses to your comments. If you have any other concerns, we would like to further discuss. If we have addressed your concerns, we sincerely hope you can reconsider your evaluation of our paper.
> > >
> > > Sincerely,
> > >
> > > Authors

---

> > > > ### Author Response · Authors · 2024-08-14
> > > > **Gentle Reminder**
> > > >
> > > > Dear Reviewer em5U,
> > > >
> > > > We are now in the **final 6 hours** of the discussion period.
> > > >
> > > > Please let us know if you have any further questions or concerns. We greatly appreciate your feedback and are committed to addressing any potential issues.
> > > >
> > > >
> > > > Sincerely,
> > > >
> > > > Authors

---

### Official Review · Reviewer_aLWE · 2024-07-13

**Soundness:** 3
**Presentation:** 3
**Contribution:** 3
**Rating:** 7
**Confidence:** 2

**Summary:**

The paper presents a comprehensive approach for dual-target drug design by reprogramming pretrained single-target diffusion models. It introduces a curated dataset derived from synergistic drug combinations, and two methods, COMPDIFF and DUALDIFF. The experimental results demonstrate the outperformance of the proposed methods compared to the baselines.

**Strengths:**

The experimental setup is rational, including multiple appropriate baselines and comprehensive evaluation metrics. In addition, this proposed method can serve as a general framework where any pretrained generative models for SBDD can be applied to dual-target drug design without fine-tuning.

**Weaknesses:**

The generalization of this framework could be discussed in more detail.

**Questions:**

None

**Limitations:**

The authors clearly discussed the limitations and potential improvements of the methods.

---

> ### Author Rebuttal · Authors · 2024-08-07
>
> Thanks for your positive feedback. Please see below for our responses to the comments.
>
> **Q1: Discussion about the generalization of this framework.**
>
> A1:
> Our proposed framework is **general** and **any types of pretrained generative models** for structure-based drug design can be applied to dual-target drug design without any fine-tuning. In our work, we use TargetDiff as a demonstrative case. Additionally, as a representative example, Pocket2Mol can also be reprogrammed. See our response to reviewer xVLn for details (Q4 & A4).
>
> Our methods can also be easily applied to **multi-target drug design**, though we focus on dual-target drug design in the experiments in our paper. Specifically, for the pocket alignment procedure, we can select a pocket and align all other pockets to it.
>
> Since we focus on reprogramming single-target diffusion models for dual-target drug design in a **zero-shot manner**, there are two different tasks and datasets for training (single-target) and inference (dual-target), which require strong generalization ability of our methods. The promising experimental results show that our reprogrammed model can effectively design drugs that bind to dual targets, demonstrating the generalization ability of our framework.

---

> ### Author Response · Authors · 2024-08-13
> **Gentle Reminder**
>
> Thanks again for your feedback! We sincerely appreciate the time and effort you have dedicated to reviewing our paper.
>
> To respond to your comments, we have discussed the generalization of the framework from various perspectives in detail.
>
> As the discussion phase will end soon, we kindly request, if possible, that you review our rebuttal at your earliest convenience. If you have any other concerns, we would like to further discuss. If we have addressed your concerns, we sincerely hope you can reconsider the evaluation of our paper.

---

> > ### Comment · Reviewer_aLWE · 2024-08-13
> >
> > I thank the authors for their thoughtful engagement and additional clarifications! I have read through the rebuttal and adjusted the overall rating.

---

> > > ### Author Response · Authors · 2024-08-14
> > >
> > > Thanks for your positive support! We will add the discussion about the generalization of the framework to our updated manuscript as you suggested.

---

### Official Review · Reviewer_xVLn · 2024-07-16

**Soundness:** 3
**Presentation:** 3
**Contribution:** 3
**Rating:** 7
**Confidence:** 4

**Summary:**

This work studies a dual-target drug design using diffusion models trained on single-target protein-ligand complex pairs.

There two proposed methods, COMPDIFF and DUALDIFF, both are aim to align and generate dual-target ligands using SE(3)-equivariant composed message passing. The paper also introduces a curated dataset based on synergistic drug combinations to evaluate the effectiveness of these methods.

In general, I like this work and its perspective from single-cell / bio-system reprogramming to drug design tasks.

On interesting point is, for pre-trained models, there are also neural model reprogramming works [A,B] and some theoretical studies on selecting pre-trained model for mis-tasks alignment in [B].

It would be attracting more audience by adding some related neural model-level reprogramming in the related works or future theoretical analysis.

***
**References**

A. Adversarial reprogramming of neural networks, ICLR 2019

B. Voice2series: Reprogramming acoustic models for time series classification, ICML 2021

C. Single-cell expression analyses during cellular reprogramming reveal an early stochastic and a late hierarchic phase. Cell 2012

**Strengths:**

- overcoming drug resistance and improving therapeutic efficacy by using diffusion models for dual-target drug design

- alignment of dual pockets using protein-ligand binding priors is interesting and novel

- The creation of the dataset based on synergistic drug combinations is a good contribution

**Weaknesses:**

- the alignment side, there are more depth to studies motivated by measurement theories or so.

-  illustrations to explain the methodology, e.g., the process of SE(3)-equivariant composed message passing would be a plus

**Questions:**

1. Would the authors provide more details on detailed justification for the choice of baselines?

2. How would be challenges on reprogramming other generative model different from diffusion based model?

**Limitations:**

- Not previous, some limitation on the wet labs perhaps.

---

> ### Author Rebuttal · Authors · 2024-08-07
>
> Thank you for your positive feedback. Please see below for our responses to the comments.
>
> **Q1: Further studies of the alignment motivated by measurement theories or so.**
>
> A1:
> The alignment of the dual pockets is dependent on the molecule selected to compute the alignment transformation (i.e., the anchor molecule that we mentioned in line 313-319). Thus, we provided ablation studies on various alignment algorithms based on different types of anchor molecules (see Section 4.3 and Table 2). The results show that the performance of various alignment methods (CompDiff/DualDiff-RMSD/Score) is consistently superior to the baseline (CompDiff/DualDiff-Center).
>
> Our paper focuses on the application, i.e., dual-target drug design instead of theory, which is out of our scope. We would like to leave it as future work.
>
> **Q2: Illustrations to explain the methodology, e.g., the process of SE(3)-equivariant composed message passing.**
>
> A2:
> We have illustrated our method in Figure 1. To be more clarified, we would add more detailed legends and captions to promote understanding. As for SE(3)-equivariant composed message passing, see the content inside the dashed box in Figure 1, where the gray / red / green nodes represent atoms of ligand molecules being generated / pocket P1 / pocket P2, respectively. The arrows in red and green represent the messages (or drifts) computed over two heterogenous knn graphs with dual targets P1 and P2, respectively. (Specifically, one graph is built upon the gray and red nodes, and the other is built upon the gray and green nodes.) Then the messages are further composed and the dark arrow represents the SE(3)-equivariant composed message. Notably, the compositional operation preserves the SE(3)-equivariance (see Appendix D for the proof). The composed message is used to update the atom positions and types of ligand molecules being generated. The complete reverse sampling process consists of a series of such operations (1000 steps in practice).
>
> **Q3: More details on detailed justification for the choice of baselines.**
>
> A3: Considering the strict page limit of the submission, we describe the choice of baselins in Section 3.3. We will elaborate on the justification in the appendix in the revision. Here, we provide more detailed justification as follows: One of the traditional strategies for dual-target drug design (refer to [1] as a comprehensive review) is linking pharmacophores by domain experts, such as chemists or pharmaceutical scientists. We use SOTA deep-learning-based linker design methods (DiffLinker [2] and LinkerNet [3]) to mimic this procedure. Since each target pocket has a corresponding ligand molecule in our curated dataset, we can break the ligand molecule into fragments and select the ones critical for binding as pharmacophores. In this case, given respective critical fragments of dual pockets, we can design linkers and resemble them into complete molecules. Ideally, the designed molecules contain critical pharmacophores that account for binding to dual targets and serve as a potential candidate for dual-target drugs.
>
> **References:**
>
> [1] Sun, Dejuan, Yuqian Zhao, Shouyue Zhang, Lan Zhang, Bo Liu, and Liang Ouyang. "Dual-target kinase drug design: Current strategies and future directions in cancer therapy." European Journal of Medicinal Chemistry 188 (2020): 112025.
>
> [2] Igashov, Ilia, Hannes Stärk, Clément Vignac, Arne Schneuing, Victor Garcia Satorras, Pascal Frossard, Max Welling, Michael Bronstein, and Bruno Correia. "Equivariant 3D-conditional diffusion model for molecular linker design." Nature Machine Intelligence (2024): 1-11.
>
> [3] Guan, Jiaqi, Xingang Peng, Peiqi Jiang, Yunan Luo, Jian Peng, and Jianzhu Ma. "LinkerNet: fragment poses and linker co-design with 3D equivariant diffusion." Advances in Neural Information Processing Systems 36 (2024).
>
> **Q4: Challenges on reprogramming other generative models different from diffusion-based models.**
>
> A4:
> As pointed in line 66-68, our method can be viewed as a general framework where any pretrained generative models for structure-based drug design (SBDD) can be applied to dual-target drug design without any fine-tuning.
>
> One example is reprogramming the autoregressive models for SBDD, e.g., Pocket2Mol [1] and GraphBP [2]. They are GNN-based methods that generate 3D molecules by sequentially placing atoms into a protein binding pocket. In our framework, the alignment procedure is independent of the type of generative models. We only need to compose the conditional distribution at each step of autoregressive sampling, which is actually straightforward and much simpler than the compositional sampling of diffusion models.
>
> We have tested the performance of Pocket2Mol for single-target drug design (see Table 4 in Appendix B). The average high affinity ratio of Pocket2Mol is only 2.1%, far behind TargetDiff (56.0%). Thus, we do not reprogram it in practice, considering its unsatisfactory performance.
>
> **References:**
>
> [1] Peng, Xingang, Shitong Luo, Jiaqi Guan, Qi Xie, Jian Peng, and Jianzhu Ma. "Pocket2mol: Efficient molecular sampling based on 3d protein pockets." In International Conference on Machine Learning, pp. 17644-17655. PMLR, 2022.
>
> [2] Liu, Meng, Youzhi Luo, Kanji Uchino, Koji Maruhashi, and Shuiwang Ji. "Generating 3d molecules for target protein binding." arXiv preprint arXiv:2204.09410 (2022).
>
> **Q5: "Some limitation on the wet labs perhaps."**
>
> A5:
> We have discussed the limitations of our work in Appendix E. We would like to validate our methods by wet-lab experiments in the future, if possible. And we will add this point in Appendix E as you suggest.

---

> > ### Comment · Reviewer_xVLn · 2024-08-11
> >
> > Thanks the authors for the comments. I think it is a baseline paper but I am looking for the impact on the acceptance side.
> >
> > On the related references, could the authors give some discussion on the connections to the neural model reprogramming as the commented list below?

---

> ### Author Response · Authors · 2024-08-11
> **Response to Reviewer xVLn**
>
> Thanks for your comments! We discuss the connections to the references of neural model reprogramming as follows:
>
> [1] proposed adversarial reprogramming. Specifically, given a model trained on a task which maps $x$ to $f(x)$, the adversary aims to repurpose the model for a new task which maps $g(\tilde{x})$ for inputs $\hat{x}$, by learning mapping functions $h_f(\cdot;\theta)$ and $h_g(\cdot;\theta)$ to make $h_g(f(h_f(\hat{x})))$ approximate $g(\hat{x})$. They successfully demonstrated their methods on image classification tasks. For example, a trained adversarial program can cause a classifier trained on ImageNet to an MNIST classifier.
>
> [2] reprograms acoustic models for time series classification, through input transformation learning and output label mapping, which are similar to $h_f(\cdot)$ and $h_g(\cdot)$, respectively. The motivation is that data scarcity hinders researchers from using large-scale deep learning models for time-series tasks while many large-scale pre-trained speech processing models are available. [3] also follows this paradigm to reprogram pretrained language models, BERT trained on English corpus, for the antibody sequence infilling task.
>
> **The above works all aim to reprogram a model trained on a task to perform a new task.** Our work also falls into this paradigm, since we focus on reprogramming models trained for single-target drug design to design dual-target drugs. And we also have a **similar motivation** of [2,3], because **there is no training data for dual-target drug design**. Differently, we focus on **reprogramming a diffusion model equipped with SE(3)-equivariance** which has not been widely explored in previous works due to its complex sampling process during inference. And, notably, all the above works require additional training to learn the additional parameters of input transformation (i.e., $h_f(\cdot;\theta)$ in [1]) for reprogramming. However, our framework works in a **zero-shot manner**, which means **we only need to modify the sampling process of the diffusion model to perform the novel task without any additional training**. Specifically, our proposed pocket alignment resembles the "input transformation", i.e., $h_f(\cdot)$, and the compositional sampling resembles the "output label mapping", i.e., $h_g(\cdot)$, but no learnable parameters are introduced for reprogramming. [2] provided theoretical results on selecting a pre-trained model for reprogramming, but it is not easy to directly apply the results to our case due to the intricate properties of diffusion models. It is interesting to study the theoretical results for our framework and we would like to leave it as a future work. Though our work focuses on dual-target drug design, a task of great significance to AI-driven drug discovery, we believe it has a broader impact beyond this field.
>
> In our work, we propose a curated dataset for dual-target drug design, strong baselines, and our highly non-trivial methods. We believe our work is more than a baseline.
>
> Thanks again for pointing out the references on neural model reprogramming, which enhances the comprehensiveness of our research work!
>
>
> **References:**
>
> [1] Adversarial reprogramming of neural networks, ICLR 2019
>
> [2] Voice2series: Reprogramming acoustic models for time series classification, ICML 2021
>
> [3] Reprogramming Pretrained Language Models for Antibody Sequence Infilling, ICML 2023

---

> ### Comment · Reviewer_xVLn · 2024-11-22
> **Camera-Ready Version Issues**
>
> I am reviewing the camera-ready version. None of the above discussion has been included in the final version. Please incorporate the additional discussion and references into the final version as per the previous discussion.

---

> > ### Comment · Area_Chair_PQVT · 2024-11-22
> > **Addressing the issues with camera ready**
> >
> > Dear Authors,
> >
> > Please address the concern raised by the Reviewer to incorporate the above discussions submitted as response in the camera ready manuscript.

---

> > > ### Public Comment · ~Xiangxin_Zhou1 · 2024-11-22
> > >
> > > Dear Area Chair PQVT,
> > >
> > > Thank you for the reminder.  We appreciate the efforts Reviewer xVLn and you have made to help make our paper more comprehensive.
> > >
> > > As per your suggestion, we have prepared a manuscript with additional references and discussions. However, since the deadline for updating the camera-ready version has passed and there is no available option to upload our revisions, we have reached out to Program Chairs at pc2024@neurips.cc with our updated manuscript attached, requesting their assistance in uploading it.
> > >
> > > Thank you once again for your kind reminder and support.
> > >
> > > Sincerely,
> > >
> > > Authors

---

> > ### Public Comment · ~Xiangxin_Zhou1 · 2024-11-22
> >
> > Dear Reviewer xVLn,
> >
> > Thank you for your reminder. We appreciate your valuable feedback and have noted the need for additional discussion and references. We will incorporate these revisions and update the camera-ready version shortly.
> >
> > Sincerely,
> >
> > Authors

---

> ### Comment · Reviewer_xVLn · 2024-11-26
>
> Hello Authors,
>
> The PCs have informed in the general email that there will be a make-up opportunity after the conference for submitting the camera-ready version.
>
> However, as a gentle reminder, the NeurIPS review process aims to enhance the quality of submissions.
>
> It is the authors’ responsibility to ensure the camera-ready version complies with the guidelines.
>
> Please ensure that the claims discussed in the rebuttal, as per each reviewer’s feedback (I saw the same for the other reviewers in this submission), have been carefully incorporated, as required during the make-up opportunity and in the future review processes.

---

> ### Public Comment · ~Xiangxin_Zhou1 · 2024-11-26
>
> Dear Reviewer xVLn,
>
> Thanks for your kind reminder.
>
> We have sent an email to the Program Chair with our new manuscript, which actually incorporates important discussions with all reviewers. Additionally, we have just updated our manuscript on arXiv, and it should be openly accessible tomorrow. You can check it there if you wish.
>
> Furthermore, we'll be attentively looking out for any additional opportunities to make updates. If such an opportunity becomes available, we will surely upload our revised version. We truly appreciate your reminder once again.
>
> Best regards,
>
> Authors

---

### Comment · Area_Chair_PQVT · 2024-08-13
**Request to reviewers for post-rebuttal feedback**

Dear Reviewers,

We are into the last 24 hours of author-reviewer discussions. As you might have noticed, authors have actively tried to engage in the discussions. Hence, I request all the reviewers to please respond to the rebuttal at the earliest so that the authors get a fair chance to represent themselves.

Thank you,

AC

---

### Decision · Program_Chairs · 2024-09-25

**Decision:**

Accept (poster)

**Comment:**

Dual-Target Drug Design is an important problem in drug discovery for which authors present a solution based on diffusion models and SE(3) equivariant GNNs. Reviewers raised some concerns regarding the reporting including computational time and efficiency, choice of baselines, need for the method in industry, and other minor concerns. Authors have satisfactorily addressed these comments. It is recommended that the additional results and new insights that came of the discussions are added in the camera ready version of the manuscript.